# Coordinated action of multiple transporters in the acquisition of essential cationic amino acids by the intracellular parasite *Toxoplasma gondii*

**Stephen J. Fairweather**[1‡]*, **Esther Rajendran**[1‡], **Martin Blume**[2,3], **Kiran Javed**[1], **Birte Steinhöfel**[1,4], **Malcolm J. McConville**[2], **Kiaran Kirk**[1], **Stefan Bröer**[1], **Giel G. van Dooren**[1]*

**1** Research School of Biology, Australian National University, Canberra, Australian Capital Territory, Australia, **2** Department of Biochemistry and Molecular Biology and the Bio21 Institute of Molecular Science and Biotechnology, University of Melbourne, Parkville, Victoria, Australia, **3** Robert Koch Institute, Berlin, Germany, **4** Humboldt University Berlin, Berlin, Germany

‡ These authors share first authorship on this work.
* stephen.fairweather@anu.edu.au (SJF); giel.vandooren@anu.edu.au (GGvD)

**Data Availability Statement:** The authors confirm that all data underlying the findings are fully available without restriction. The mass

## Abstract

Intracellular parasites of the phylum Apicomplexa are dependent on the scavenging of essential amino acids from their hosts. We previously identified a large family of apicomplexan-specific plasma membrane-localized amino acid transporters, the ApiATs, and showed that the *Toxoplasma gondii* transporter *Tg*ApiAT1 functions in the selective uptake of arginine. *Tg*ApiAT1 is essential for parasite virulence, but dispensable for parasite growth in medium containing high concentrations of arginine, indicating the presence of at least one other arginine transporter. Here we identify *Tg*ApiAT6-1 as the second arginine transporter. Using a combination of parasite assays and heterologous characterisation of *Tg*ApiAT6-1 in *Xenopus laevis* oocytes, we demonstrate that *Tg*ApiAT6-1 is a general cationic amino acid transporter that mediates both the high-affinity uptake of lysine and the low-affinity uptake of arginine. *Tg*ApiAT6-1 is the primary lysine transporter in the disease-causing tachyzoite stage of *T. gondii* and is essential for parasite proliferation. We demonstrate that the uptake of cationic amino acids by *Tg*ApiAT6-1 is 'trans-stimulated' by cationic and neutral amino acids and is likely promoted by an inwardly negative membrane potential. These findings demonstrate that *T. gondii* has evolved overlapping transport mechanisms for the uptake of essential cationic amino acids, and we draw together our findings into a comprehensive model that highlights the finely-tuned, regulated processes that mediate cationic amino acid scavenging by these intracellular parasites.

## Author summary

The causative agent of toxoplasmosis, *Toxoplasma gondii*, is a versatile intracellular parasite that can proliferate within nucleated cells of warm-blooded organisms. In order to

spectrometry-based metabolomics data have been uploaded to the Mendeley Data repository, with the GC-MS data from Fig 2A accessible at https://data.mendeley.com/datasets/64tdwbccx9/1 and the LC-MS data in Fig 6 accessible at https://data.mendeley.com/datasets/893nsf2bnc/1 (data set 1), https://data.mendeley.com/datasets/xsb56m9gs8/1 (data set 2), and https://data.mendeley.com/datasets/4xy7zppt2s/1 (data set 3).

**Funding:** This work was supported by Discovery Grants from the Australian Research Council to KK, GGvD and SB (DP150102883) and to GGvD and KK (DP200100483). MJM is a NHMRC Principal Research Fellow (APP1154540). The funders had no role in study design, data collection and analysis, decision to publish, or preparation of the manuscript.

**Competing interests:** The authors have declared that no competing interests exist.

survive, *T. gondii* parasites must scavenge the cationic amino acids lysine and arginine from their hosts. In a previous study, we demonstrated that a plasma membrane-localized protein called *Tg*ApiAT1 facilitates the uptake of arginine into the parasite. We found that parasites lacking *Tg*ApiAT1 could proliferate when cultured in medium containing high concentrations of arginine, suggesting the existence of an additional uptake pathway for arginine. In the present study, we demonstrate that this second uptake pathway is mediated by *Tg*ApiAT6-1, a protein belonging to the same solute transporter family as *Tg*ApiAT1. We show that *Tg*ApiAT6-1 is the major lysine transporter of the parasite, and that it is critical for parasite proliferation. Furthermore, we demonstrate that *Tg*ApiAT6-1 can transport arginine into parasites under conditions in which arginine concentrations are high and lysine concentrations are comparatively lower. These data support a model for the finely-tuned acquisition of essential cationic amino acids that involves multiple transporters, and which likely contributes to these parasites being able to survive and proliferate within a wide variety of host cell types.

## Introduction

Intracellular parasites of the phylum Apicomplexa are the causative agents of a diverse range of diseases in humans and domestic livestock, imposing major health and economic burdens in many countries. The apicomplexan parasite *Toxoplasma gondii* infects up to one-third of the human population, and is the causative agent of the disease toxoplasmosis. Although usually asymptomatic in healthy adults, toxoplasmosis can cause lethal encephalitis in immunocompromised patients. In addition, there are ~200,000 cases of congenital toxoplasmosis worldwide per year, resulting in a range of birth defects including microencephaly, anemia, vision loss, premature birth and stillborn infants [1–3].

The evolution of apicomplexan parasites from free-living ancestors has been associated with the loss of numerous metabolic pathways and increased dependence on the host for essential carbon sources and nutrients [4,5]. To allow for the loss of these pathways, parasites have co-evolved new mechanisms to acquire nutrients from their hosts [4,5]. As they proceed through their multi-stage life-cycles [6–8], apicomplexans adapt to a range of physiological and biochemical environments. *T. gondii* is an exemplar of this successful adaptation. It is thought to be able to infect all nucleated cells in all warm-blooded animals [9–12], indicating a remarkable ability to acquire essential nutrients in nutritionally diverse niches [4]. Like other apicomplexans [6,13–15], *T. gondii* is auxotrophic for numerous amino acids and other amino compounds [9,11,16]. Arginine (Arg) and its downstream metabolic products ornithine and polyamines cannot be synthesized *de novo* and must be acquired from the host [17,18]. Leucine (Leu), isoleucine (Ile), valine (Val), methionine (Met), threonine (Thr), and histidine (His) must also be acquired from the host [11]. Despite initial indications that *T. gondii* could synthesise aromatic amino acids utilising the shikimate pathway [11,19], the disease-causing tachyzoite stage of the parasite has been shown to be auxotrophic for tyrosine (Tyr), tryptophan (Trp), and phenylalanine (Phe) [20–23]. While lysine (Lys), or its direct metabolic precursor L-2-aminoadipate-6-semialdehyde, are predicted to be essential, this requirement has not been tested *in situ* [24]. As a result of these auxotrophies, *T. gondii* is reliant primarily on multiple plasma membrane-localized transporters to salvage amino acids from their hosts [9], although the endocytosis and lysosomal degradation of proteins may also play a role [25].

Annotation of the genome of the malaria-causing parasite *Plasmodium falciparum* revealed the presence of a hypothetical family of solute transporters [26] which were subsequently

shown to transport amino acids and termed Apicomplexan Amino acid Transporters (ApiATs) [20, 27,28]. Phylogenetic analysis using Hidden Markov Model (HMM) profiles demonstrated that this divergent protein family is most closely related to the neutral amino acid-transporting LAT3 family of Major Facilitator Superfamily transporters [20] (SLC43 in mammals). To date, three ApiAT proteins have been characterized and shown to transport essential amino acids across the plasma membrane. In *T. gondii*, *Tg*ApiAT1 (previously known as *Tg*NPT1; [27]; www.toxodb.org gene identifier TGME49_215490) and *Tg*ApiAT5-3 [20,28] are Arg and aromatic/neutral amino acid transporters, respectively, while in *Plasmodium berghei*, *Pb*ApiAT8 (previously termed *Pb*NPT1) is a general cationic amino acid transporter [27]. All three ApiATs have been shown to be important at particular parasite life-cycle stages: the *T. gondii* transporters for tachyzoite proliferation [20,27], and *Pb*ApiAT8 for *P. berghei* gamete development in mice and transmission to *Anopheles* mosquitos [27,29,30]. *Tg*ApiAT1 expression is regulated by the intracellular availability of Arg, allowing parasites to exert tight control over the uptake of this amino acid [31]. The dearth of other candidate amino acid transporter homologues in apicomplexan genomes suggests that ApiATs may be the primary amino acid/ amino acid metabolite transporter family in these parasites [20,31,32].

In *T. gondii*, the ApiAT family has undergone expansive radiation to 16 members [20]. All ApiATs characterized to date have been shown to be equilibrative transporters, facilitating the transmembrane passage of their amino acid substrates without the direct involvement of any other co-substrates (*e.g.* ions such as $Na^+$ or $H^+$). *Tg*ApiAT5-3 has also been shown to facilitate the bi-directional exchange of its aromatic/neutral amino acid substrates [20].

In addition to *Tg*ApiAT1 and *Tg*ApiAT5-3, two other *T. gondii* ApiAT proteins, *Tg*ApiAT2 and *Tg*ApiAT6-1, have been shown to be important for tachyzoite proliferation [20,33]. Specifically, genetic disruption of *Tg*ApiAT2 is associated with reduced parasite growth *in vitro*, an effect that was exacerbated in a minimal amino acid medium [20], while *Tg*ApiAT6-1 was unable to be genetically disrupted, consistent with it being essential for tachyzoite proliferation [20]. In our previous characterisation of *Tg*ApiAT1, we noted that *Tg*ApiAT1 could be knocked out if parasites were cultured in medium containing >five-fold higher concentrations of Arg than Lys [27], suggesting the presence of a second cationic amino acid transporter system that may regulate uptake of both Lys and Arg [27].

Here, we identify *Tg*ApiAT6-1 as this second transporter. We demonstrate that *Tg*ApiAT6-1 has a dual physiological role in the parasite, functioning as the essential high-affinity Lys transporter in addition to being a lower-affinity Arg transporter that mediates Arg uptake under arginine-replete conditions. We elucidate the transport mechanism of *Tg*ApiAT6-1, as well as that of *Tg*ApiAT1, showing both to be bidirectional uniporters with the capacity to mediate amino acid exchange, and the capacity to facilitate the intracellular accumulation of these two essential cationic amino acids. We integrate our findings into a detailed model of cationic amino acid ($AA^+$) uptake in *T. gondii*, in which the parasite is exquisitely adapted to ensure coordinated acquisition of these essential nutrients.

## Results

### *Tg*ApiAT6-1 is important for parasite proliferation

In a previous study of the ApiAT family in *T. gondii*, we demonstrated that *Tg*ApiAT6-1 localized to the plasma membrane of the parasite [20]. We were unable to disrupt the *Tg*ApiAT6-1 locus using CRISPR/Cas9 genome editing, suggesting that *Tg*ApiAT6-1 (www.toxodb.org gene identifier TGME49_240810) may be essential for proliferation of the tachyzoite stage [20]. To test this hypothesis, and to facilitate subsequent functional characterisation of *Tg*ApiAT6-1, we generated a regulatable parasite strain, r*Tg*ApiAT6-1, in which *Tg*ApiAT6-1

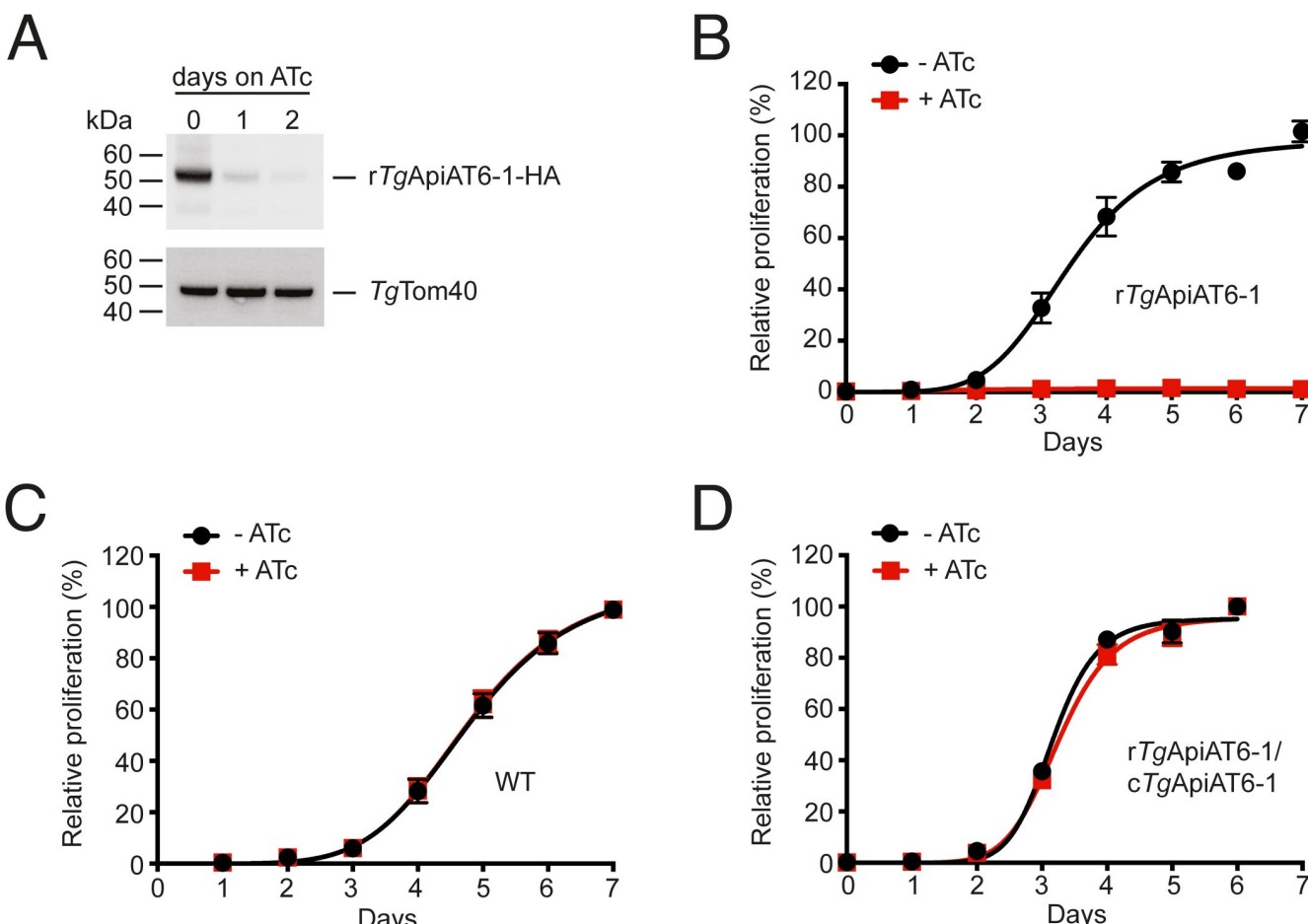

**Fig 1. *Tg*ApiAT6-1 is important for parasite proliferation. A**. Western blot analysis of proteins extracted from r*Tg*ApiAT6-1-HA parasites cultured in the presence of ATc for 0–2 days, and detected with antibodies against HA or *Tg*Tom40 (as a loading control). **B-D**. Fluorescence growth assays measuring the proliferation of r*Tg*ApiAT6-1 parasites (B), WT parasites (C), or r*Tg*ApiAT6-1 parasites complemented with a constitutively-expressed copy of *Tg*ApiAT6-1 (r*Tg*ApiAT6-1/c*Tg*ApiAT6-1; D). Parasites were cultured for 6 or 7 days in the absence (black) or presence (red) of ATc. Parasite proliferation is expressed as a percentage of parasite proliferation in the -ATc condition on the final day of the experiment for each strain. Data are averaged from 3 technical replicates (± S.D.) and are representative of three independent experiments.

expression could be knocked down through the addition of anhydrotetracycline (ATc) (S1 Fig). A 3' hemagglutinin (HA) epitope tag was introduced into the r*Tg*ApiAT6-1 locus and knockdown of the *Tg*ApiAT6-1 protein was observed after 2 days cultivation in the presence of ATc (Fig 1A). The r*Tg*ApiAT6-1 strain expressed a tdTomato transgene to allow measurement of growth in the absence or presence of ATc for 7 days using fluorescence proliferation assays, as previously described [27]. Compared to parasites cultured in the absence of ATc, we observed a major defect in proliferation in r*Tg*ApiAT6-1 parasites cultured in the presence of ATc (conditions under which *Tg*ApiAT6-1 is knocked down; Fig 1B). ATc had no effect on the proliferation of wild type (WT) parasites under the same conditions (Fig 1C). These data indicate that the knockdown of r*Tg*ApiAT6-1 is associated with a severe impairment of parasite proliferation.

To determine whether the proliferation phenotype we observed was due solely to depletion of *Tg*ApiAT6-1, we introduced a constitutively-expressed copy of *Tg*ApiAT6-1 into r*Tg*ApiAT6-1 parasites, generating the strain r*Tg*ApiAT6-1/c*Tg*ApiAT6-1. The presence of the constitutively-expressed *Tg*ApiAT6-1 fully restored parasite proliferation in the presence of ATc

(Fig 1D). Together, these data indicate that *Tg*ApiAT6-1 is important for proliferation of the tachyzoite stage of *T. gondii* parasites.

## *Tg*ApiAT6-1 is a high affinity lysine transporter

In order to investigate whether r*Tg*ApiAT6-1 is an amino acid transporter, we incubated r*Tg*ApiAT6-1 parasites grown for two days in the absence or presence of ATc in amino acid-free RPMI 1640 medium supplemented with 2 mg/ml of a [13C]-labelled amino acid mix. Parasites were incubated in the [13C]-labelled amino acid mix for 15 mins, then polar metabolites were extracted and analysed by GC-MS. We compared the fractional abundance of 13C-labelled amino acids to the total abundance of each amino acid following the 15 min uptake period (Fig 2A). Of the 17 amino acids detected by GC-MS, only the uptake of 13C-Lys was significantly reduced when *Tg*ApiAT6-1 expression was knocked down. These findings were replicated in a recent study that adopted a similar approach in measuring the uptake of 13C-labelled amino acids in parasites depleted in *Tg*ApiAT6-1 [34]. These data suggested that *Tg*ApiAT6-1 may be a Lys transporter, although it could also mediate the uptake of other amino acids not detected under the transport conditions of the experiments, or not detected by GC-MS, such as Arg. Arg is readily converted into L-ornithine during in-line heating and separation for GC-MS such that very little remains chemically unmodified [35,36].

To characterise further the substrate specificity of *Tg*ApiAT6-1, and to investigate the transport mechanism, we expressed *Tg*ApiAT6-1 in *Xenopus laevis* oocytes. We demonstrated that *Tg*ApiAT6-1 was expressed on the plasma membrane of oocytes, where it is detectable at two distinct molecular masses: one at ~48 kDa, consistent with a monomer of *Tg*ApiAT6-1, and another at ~95 kDa, which may represent a dimeric form of the protein (S2A Fig). After optimising its expression in oocytes (S2B and S2C Fig), we investigated the substrate specificity of *Tg*ApiAT6-1. We measured the uptake of a range of radiolabelled amino acids and amino acid derivatives in *Tg*ApiAT6-1-expressing oocytes, a selection of which are shown in Fig 2B. Consistent with the metabolomics data, *Tg*ApiAT6-1 mediated Lys uptake (Fig 2B). Notably, *Tg*ApiAT6-1 also mediated uptake of Arg and some neutral amino acids including Met and Leu (Fig 2B).

In the experiment measuring the uptake of [13C]-labelled amino acids into parasites, neither Met nor Leu showed discernible changes in uptake following knock down of *Tg*ApiAT6-1 (Fig 2A). This may be because *Tg*ApiAT6-1 has a higher affinity for Lys than for the neutral amino acids, such that under the conditions of the 13C-labelled amino acid uptake experiment, the Lys in the medium excluded the other amino acids from the active site of the transporter. To test whether this was the case, we measured *Tg*ApiAT6-1-mediated uptake of Arg in oocytes in the presence of a 10-fold (Fig 2C) or 100-fold (Fig 2D) higher concentration of other, unlabelled amino acids. At a 10-fold higher concentration of the unlabelled amino acid, only Lys inhibited Arg uptake (Fig 2C); however, at 100-fold higher concentrations, numerous neutral amino acids including Met, Leu, Phe and His partially inhibited Arg uptake (Fig 2D). This is consistent with the transporter having a higher affinity for Lys than for the other unlabelled amino acids tested.

To test the affinity of *Tg*ApiAT6-1 for Lys and Arg, we measured the uptake kinetics of these amino acids. The rate of substrate uptake for both Lys and Arg into oocytes expressing *Tg*ApiAT6-1 remained constant throughout the first 10 min of uptake reactions (S2D Fig) and subsequent experiments were performed within this timeframe. We found that *Tg*ApiAT6-1 has a much higher affinity for Lys than for Arg ($K_{0.5}$ for Lys was 22.8 μM ± 2.9 μM; $K_{0.5}$ for Arg was 748 μM ± 260 μM; Fig 2E and 2F and Table 1), although the maximal rate ($V_{max}$) of Arg uptake (169 ± 8 pmol/10 min/oocyte; Fig 2E and 2F and Table 1) greatly exceeded that for

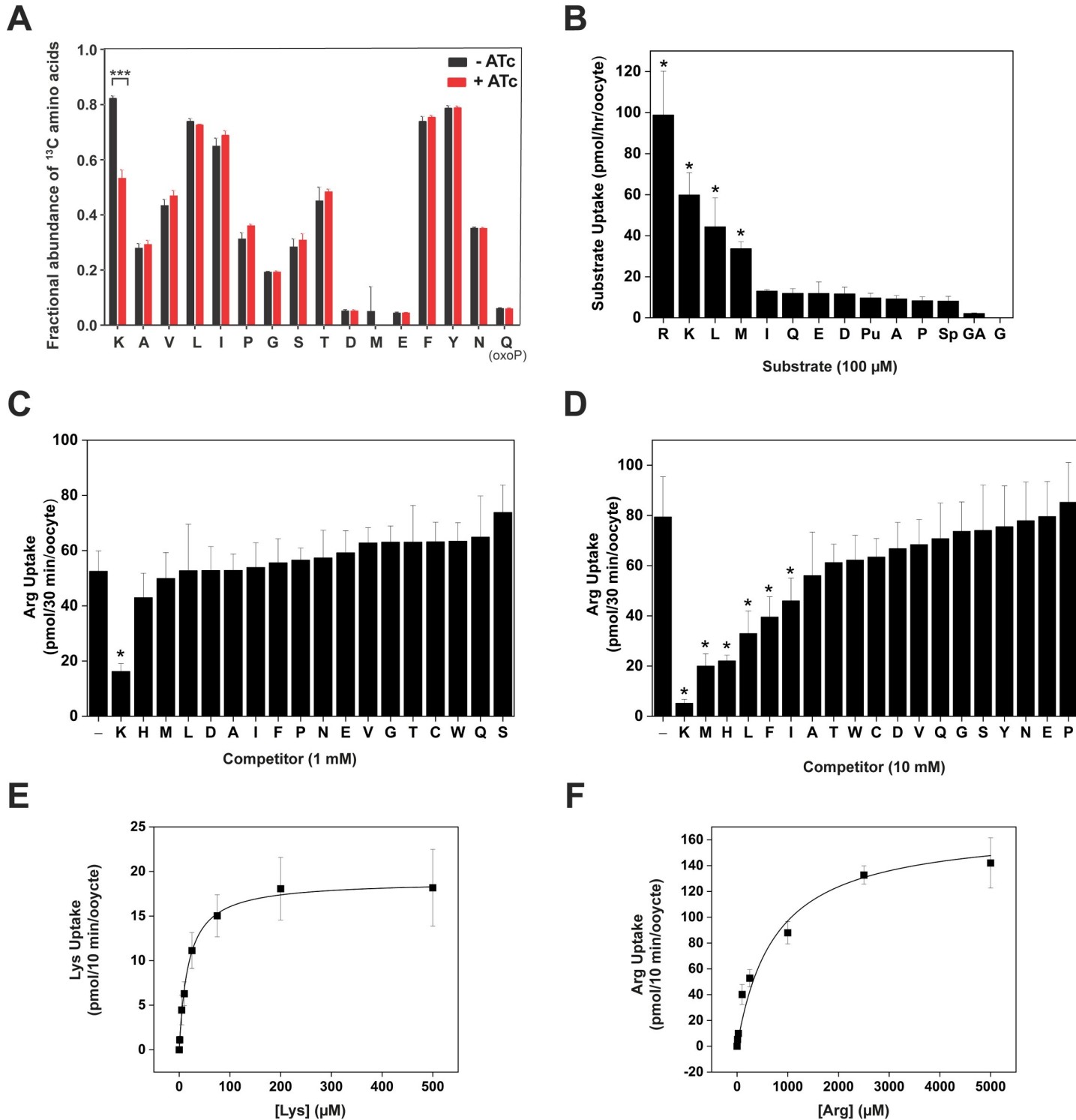

**Fig 2. *Tg*ApiAT6-1 is a cationic and neutral amino acid transporter with high affinity for Lysine. A**. Analysis of [¹³C] amino acid uptake in parasites expressing or lacking *Tg*ApiAT6-1. r*Tg*ApiAT6-1 parasites were cultured for 2 days in the absence (black) or presence (red) of ATc until natural egress, then incubated in medium containing [¹³C]-L-amino acids for 15 min. Metabolites were extracted and the fractions of [¹³C]-L-amino acids determined by GC-MS. Amino acids are represented by single letter codes; OxoP, 5-oxoproline. The data represent the mean ± S.D of 3 replicate experiments (*, *P* < 0.001, Student's *t* test. Where significance values are not shown, differences were not significant; *P* > 0.05). **B**. Uptake of a range of amino acids into oocytes expressing *Tg*ApiAT6-1. Uptake was measured in the presence of 100 μM unlabelled substrate and 1.0 μCi/ml [³H] or [¹⁴C] substrate. Amino acid substrates are represented by single letter codes, while for other metabolites: Pu, putrescine; Sp, spermidine; and GA, γ-amino butyric acid (GABA). Each bar represents the mean ± S.D. uptake of 10 oocytes for a single experiment, and each is representative of three independent experiments. The uptake

into uninjected oocytes (shown in S3A Fig) was subtracted for all substrates tested. Statistical analysis compares injected oocyte uptake to uninjected oocyte uptake for the same substrate (*$P < 0.05$, one-way ANOVA, Dunnett's post-hoc test). **C-D.** Inhibition of Arg uptake into *Tg*ApiAT6-1-expressing oocytes by a range of amino acids. Uptake of 100 μM unlabelled Arg and 1.0 μCi/ml [$^{14}$C]Arg was measured in the presence of 1 mM (C) or 10 mM (D) of the competing amino acid. Amino acid substrates are represented by single letter codes. Each bar represents the mean ± S.D. uptake of 10 oocytes for a single experiment, and are representative of three independent experiments. The first bar in each graph represents the Arg-only uptake control. The uptake in uninjected oocytes (shown in S3B and S3C Fig for the 1 mM and 10 mM competition experiments, respectively) has been subtracted for all conditions. Statistical analysis compares all bars to the Arg uptake control (*, $P < 0.05$, one-way ANOVA, Dunnett's post-hoc test). **E-F.** Steady-state kinetic analysis of Lys (E) and Arg (F) uptake into *Tg*ApiAT6-1-expressing oocytes. Uptake was measured at a range of concentrations of unlabelled Lys (E) or Arg (F) as indicated on the x-axis and 1.0 μCi/ml [$^{14}$C]Arg or [$^{14}$C]Lys. Each data point represents the mean ± S.D. uptake of 10 oocytes for a single experiment, and are representative of three independent experiments. The uptake into uninjected oocytes has been subtracted for all substrate concentrations tested.

Lys (25.4 ± 1.5 pmol/10 min/oocyte; Fig 2E and 2F and Table 1). Further evidence for the preference of *Tg*ApiAT6-1 for Lys over Arg was obtained by calculating the apparent specificity constant ($k_{cat}/ K_{0.5}$) for both substrates, which was ~five-fold greater for Lys uptake than Arg (Table 1).

Our previous study of the selective Arg transporter *Tg*ApiAT1 indicated that this transporter is electrogenic (*i.e.* transport of Arg across the membrane is coupled to a net movement of an elemental charge), and that the electrogenicity of *Tg*ApiAT1-mediated Arg transport is due to the positive charge of Arg at neutral pH [27]. We investigated whether *Tg*ApiAT6-1 is also electrogenic. When *Tg*ApiAT6-1-expressing oocytes were voltage clamped to −50 mV and perfused with 1 mM Arg, an inward current was observed (Fig 3A), reminiscent of the Arg-induced currents observed previously for *Tg*ApiAT1 [27] and other eukaryotic AA$^+$ transporters [37,38]. The first phase of these currents is represented by a steep inward current peaking at approximately –50 nA, before the second phase shows a gradual relaxation towards an equilibrated inward current of approximately –20 nA (Fig 3A). On removal (washout) of Arg from the medium, the current showed an overshoot, increasing to beyond the pre-substrate perfusion baseline current (Fig 3A), with the magnitude of this overshoot increasing with the duration of the 1 mM Arg perfusion (Fig 3B). The biphasic current pattern disappears when *Tg*ApiAT6-1 expressing, voltage-clamped oocytes were pre-injected with 1 mM Arg (Fig 3C). Together, these data can be explained by *Tg*ApiAT6-1 facilitating the bi-directional transport of Arg (*i.e.* into and out of the oocyte). In this scenario, the biphasic current and overshoot observed in oocytes reflect the movement of charge out of the oocyte as the intracellular concentration of Arg increases following uptake, something that is not observed in Arg-injected oocytes, in which the intracellular Arg concentration is high from the beginning of the experiment, and from which Arg efflux is occurring throughout.

To determine whether the observed currents are a direct consequence of the movement of Arg, or whether inorganic ions contribute to the current, we examined Arg-induced currents in voltage-clamped oocytes (−50 mV) in buffers with different salt compositions. Arg-stimulated currents gave similar values independent of salt composition (S4A and S4B Fig), consistent with Arg being the current-generating ion. Furthermore, the unidirectional *Tg*ApiAT6-1-mediated uptake of Arg into oocytes was unaffected by the absence of Na$^+$, K$^+$, Cl$^-$, Mg$^{2+}$ or

**Table 1. *Tg*ApiAT6-1 steady-state kinetic parameters for Lys and Arg.**

| Substrate | $K_{0.5}$* (μM)[†] | $V_{max}$ (pmol 10 min$^{-1}$ · oocyte)[†] | $k_{cat}$ (s$^{-1}$) [‡] | $k_{cat}/ K_{0.5}$ (M$^{-1}$s$^{-1}$)[‡] |
|---|---|---|---|---|
| Lys | 22.8 ± 2.9 | 25.4 ± 1.5 | 0.042 ± 0.004 | $1.84 \times 10^3$ |
| Arg | 748 ± 260 | 169 ± 8 | 0.28 ± 0.10 | $3.79 \times 10^2$ |

[*] Michaelis constant, the substrate concentration at half the maximal velocity of the transporter.

[†] Mean ± S.D. (*e* = 3).

[‡] relative (per oocyte).

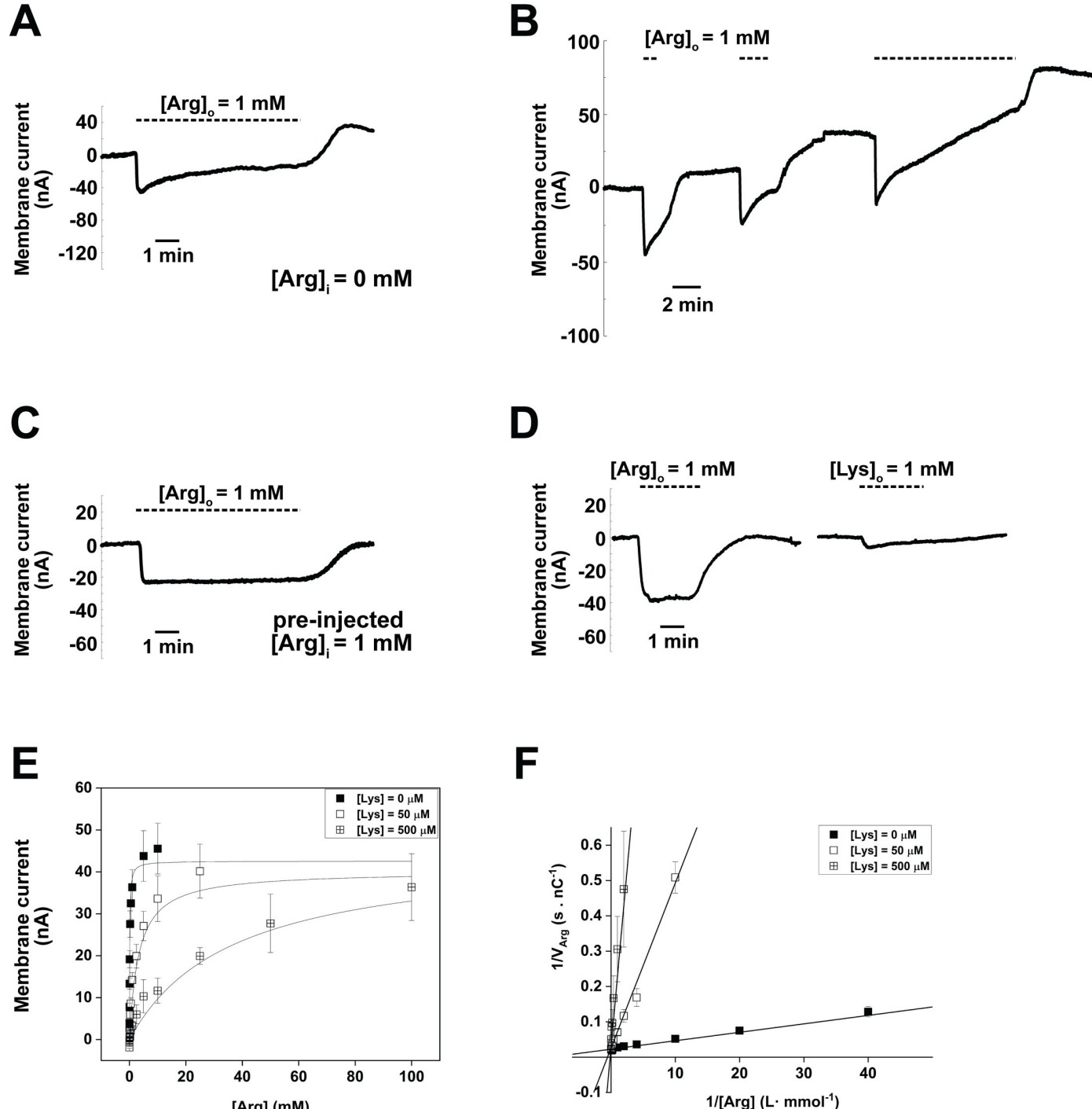

**Fig 3. Arg and Lys compete for the same binding site in *Tg*ApiAT6-1.** Electrophysiology measurements in *Tg*ApiAT6-1 expressing oocytes. All currents were recorded in two-voltage clamp configuration to record membrane current. All oocytes were voltage clamped at −50 mV and the application of substrate (Arg or Lys) are indicated by the dashed horizontal lines above the tracings. Representative current tracings were normalised to 0 nA to remove background (non-substrate induced) current. **A**. Representative current tracing of *Tg*ApiAT6-1 expressing oocytes upon the addition and subsequent washout of 1 mM extracellular Arg ($[Arg]_o$) with no pre-injection of substrate. The perfusion buffer used was ND96 (pH 7.3). **B**. Representative current tracing of a *Tg*ApiAT6-1 expressing oocyte repeatedly pulsed with 1 mM Arg for 1 min, 2 min, and 10 min with 5 min gaps in between pulses. The perfusion buffer used was ND96 (pH 7.3). **C**. Representative current tracing of a *Tg*ApiAT6-1 expressing oocyte pre-injected with 1 mM Arg ($[Arg]_i$ = 1 mM) upon the addition and subsequent washout of 1 mM extracellular Arg ($[Arg]_o$). The perfusion buffer used was ND96 (pH 7.3). **D**. Representative current tracings of *Tg*ApiAT6-1 expressing oocytes upon the addition and subsequent washout of either 1 mM extracellular Arg ($[Arg]_o$) or Lys ($[Lys]_o$) with no pre-injection of substrate. The perfusion buffer used was ND96 (pH 7.3). **E-F.** Inhibition kinetics of increasing [Lys] on Arg-induced inward currents in *Tg*ApiAT6-1 expressing oocytes. All oocytes were voltage-clamped at −50 mV and

exposed to a range of Arg concentrations at one of 3 concentrations of Lys (0, 50, 500 μM) until a new steady-state current baseline was achieved. Each data point represents the mean ± S.D. steady-state inward current (n = 14 oocytes). All 3 inhibitory [Lys] plots were fitted to either the Michaelis-Menten equation (E) with $R^2$ values of 0.97 (0 mM Lys), 0.84 (50 μM), 0.63 (500 μM), or a Lineweaver-Burke linear regression (F) with $R^2$ values of 0.94 (0 mM Lys), 0.93 (50 μM), and 0.67 (500 μM).

$Ca^{2+}$ from the medium (S4C Fig), consistent with these ions not being required for co-transport with the cationic amino acid.

We also measured Lys-induced currents in voltage-clamped *Tg*ApiAT6-1 expressing oocytes, and observed small currents of ~1–2 nA above background (Fig 3D), consistent with the low maximum velocity of *Tg*ApiAT6-1 in transporting Lys (Fig 2E and Table 1). The small relative magnitude of the Lys-mediated currents in our set-up precluded the use of electrophysiology to characterise Lys transport. Together, our electrophysiology studies demonstrate that the currents observed for *Tg*ApiAT6-1 are carried by $AA^+$ substrates of the transporter and not by any other biologically relevant ion.

Our earlier data indicated that Lys can inhibit Arg uptake into oocytes (Fig 2C and 2D). We therefore investigated whether Arg and Lys compete for the same binding site of the *Tg*ApiAT6-1 transporter. To do this, we exploited the observation that Arg, but not Lys, induces appreciable currents in voltage-clamped oocytes expressing *Tg*ApiAT6-1 (Fig 3D). We measured the steady-state kinetics of Arg-induced currents in the presence of increasing concentrations of Lys. Lys acted as a high affinity competitive inhibitor of Arg, with $K_{0.5}$ values for Arg increasing from 0.19 mM ± 0.05 mM at 0 mM Lys, to 2.3 mM ± 0.2 mM at 50 μM Lys, to 28 mM ± 8 mM at 500 μM Lys (Fig 3E). By contrast, the $V_{max}$ of Arg-induced currents remained constant at approximately −40 nA over the range of Lys concentrations tested. The changes in $K_{0.5}$ are readily observed as a change in the slope of the Lineweaver-Burke linear regressions, while the intersection of the three regression lines at the ordinate ($1/V_{Arg}$) indicates similar $V_{max}$ values (Fig 3F). These data are consistent with Arg and Lys binding to the same binding site of *Tg*ApiAT6-1, and with these substrates competing for transport by this protein. The higher affinity of *Tg*ApiAT6-1 for Lys compared to Arg means that, in a physiological setting, the contribution of *Tg*ApiAT6-1 to Arg uptake will vary with the concentration of Lys, increasing as the [Arg]:[Lys] ratio increases. We demonstrated in a previous study that Lys uptake by the parasite can be inhibited by Arg, and that Arg uptake in parasites lacking the selective Arg transporter *Tg*ApiAT1 is inhibited by Lys [27]. This is consistent with the competition between these substrates for uptake by *Tg*ApiAT6-1 that we observed in the oocyte experiments (Fig 3E and 3F).

To test whether *Tg*ApiAT6-1 contributes to Lys uptake in parasites, we measured the uptake of Lys in *Tg*ApiAT6-1 parasites cultured in the absence or presence of ATc for 2 days. We observed a significant ~16-fold decrease in the initial rate of Lys uptake upon *Tg*ApiAT6-1 knockdown ($P < 0.01$; ANOVA; Figs 4A and S5A). We next investigated the contribution of *Tg*ApiAT6-1 to Arg uptake. We measured the uptake of [$^{14}$C]Arg in *Tg*ApiAT6-1 parasites cultured in the absence or presence of ATc for 2 days. We observed a significant ~five-fold decrease in the initial rate of Arg uptake upon *Tg*ApiAT6-1 knockdown ($P < 0.01$; ANOVA; Figs 4B and S5B). These data are consistent with *Tg*ApiAT6-1 mediating the uptake of both Lys and Arg into the parasite. Neither Lys nor Arg uptake was impaired in WT parasites cultured in the presence of ATc (Figs 4A and 4B and S5C and S5D). Likewise, uptake of 2-deoxyglucose, a glucose analogue, was unaffected upon *Tg*ApiAT6-1 knockdown (S5E Fig). These data indicate that the observed defects in Lys and Arg uptake in the r*Tg*ApiAT6-1 strain were not the result of ATc addition, or of a general impairment of solute uptake or parasite viability.

We have previously shown that parasites lacking the selective Arg transporter *Tg*ApiAT1 could proliferate in medium containing high concentrations of Arg relative to Lys [27]. This

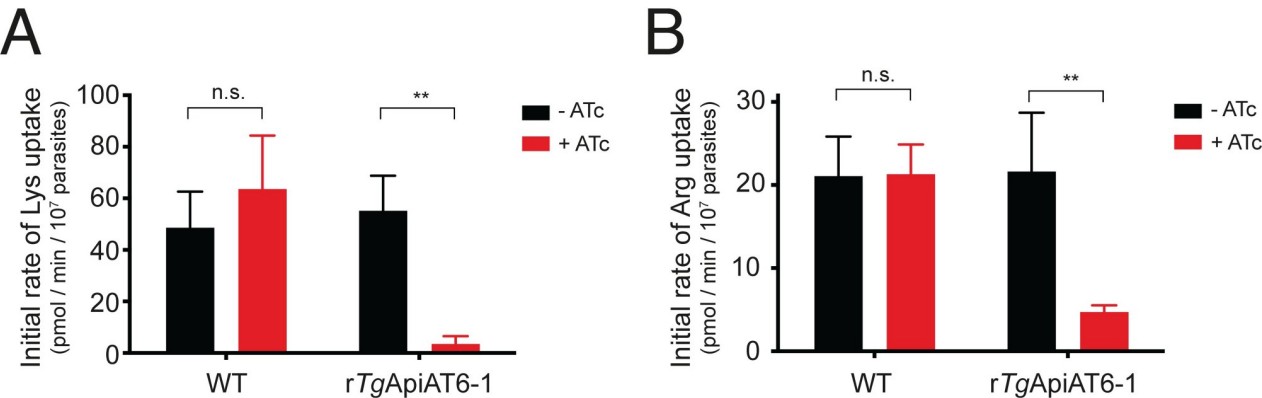

**Fig 4. *Tg*ApiAT6-1 mediates uptake of Lys and Arg in *T. gondii* parasites. A-B**. Initial rate of Lys (A) and Arg (B) uptake in WT and r*Tg*ApiAT6-1 parasites cultured in DME in the absence (black) or presence (red) of ATc for 2 days. Uptake was measured in 50 μM unlabelled Lys and 0.1 μCi/ml [$^{14}$C]Lys (**A**) or 80 μM unlabelled Arg and 0.1 μCi/ml [$^{14}$C]Arg (**B**). Initial rates were calculated from fitted curves obtained in time-course uptake experiments (S5 Fig). Data represent the mean initial uptake rate ± S.D. from three independent experiments.($^{**}$, $P = 0.01$; n.s. not significant; ANOVA with Sidak's multiple comparisons test).

suggested the presence of a second Arg transporter, which our data now indicate is *Tg*ApiAT6-1. To test whether the proliferation of parasites lacking *Tg*ApiAT6-1 was similarly dependent on the concentrations of Arg or Lys in the culture medium, we cultured r*Tg*ApiAT6-1 parasites in the absence or presence of ATc and at a range of [Lys] or [Arg]. In conditions in which *Tg*ApiAT6-1 is expressed (-ATc), we observed defects in parasite proliferation at [Lys] or [Arg] below ~50 μM (S6 Fig), consistent with previous data indicating that the parasite is auxotrophic for these amino acids [27]. In the absence of *Tg*ApiAT6-1 (+ATc), we observed minimal parasite proliferation regardless of the concentration of either Lys or Arg (S6 Fig). This indicates that, unlike *Tg*ApiAT1, defects in the proliferation of parasites lacking *Tg*ApiAT6-1 cannot be rescued by modifying the concentration of its substrates in the culture medium.

The genome of *T. gondii* parasites encodes homologues of at least some enzymes that function in the so-called diaminopimelate pathway for Lys biosynthesis [24,39]. It is conceivable, therefore, that parasites can compensate for the loss of *Tg*ApiAT6-1 by synthesising Lys via this pathway. The final putative enzyme in this pathway is termed *Tg*LysA (diaminopimelate decarboxylase; TGME49_278740). To characterise the importance of the Lys biosynthesis pathway in *T. gondii* tachyzoites, we disrupted the *Tg*LysA genomic locus in RHΔ*hxgprt*/ Tomato strain parasites using a CRISPR/Cas9-based genome editing approach. The resultant 2 bp deletion in the *Tg*LysA locus causes a frameshift mutation in the *Tg*LysA open reading frame, leading to the predicted production of a truncated *Tg*LysA protein lacking residues 281 to 686, and loss of key active site residues that are predicted to render the enzyme non-functional (S7A Fig; [40]). We termed this strain *lysA*$^{Δ281-686}$. To test the importance of *Tg*LysA for parasite proliferation, we performed fluorescence proliferation assays on WT or *lysA*$^{Δ281-68}$ parasites cultured in medium containing 0–800 μM [Lys]. As demonstrated previously ([27], S6A Fig), proliferation of WT parasites is inhibited at [Lys] below ~50 μM (S7B Fig). *lysA*$^{Δ281-68}$ parasites exhibit a similar inhibition of proliferation at [Lys] below 50 μM, and a virtually identical growth response across the entire range of tested [Lys] (S7B Fig). The observed dispensability of *Tg*LysA for tachyzoite proliferation is consistent with data from an earlier CRISPR-based screen that examined the importance of all genes encoded in the parasite genome [33].

Together, the data in S6 and S7 Figs indicate that: a) *T. gondii* tachyzoites are auxotrophic for Lys; b) proliferation of parasites lacking *Tg*ApiAT6-1 cannot be rescued by increasing the concentrations of Lys or Arg; and c) the putative Lys biosynthesis pathway in these parasites is either not functional or is unable to supply the Lys requirements needed for parasite viability. Given that the Arg transporter *Tg*ApiAT1 is still expressed when *Tg*ApiAT6-1 is depleted [31], it is unlikely that the growth defects observed upon *Tg*ApiAT6-1 depletion result solely from defects in Arg uptake. These data therefore support the hypothesis that *Tg*ApiAT6-1 is essential because it is required for the uptake of Lys. In summary, our data are consistent with the hypothesis that *Tg*ApiAT6-1 is the primary and essential Lys uptake pathway into tachyzoite-stage parasites, as well as having a role in the uptake of Arg.

## Mechanisms of cationic amino acid acquisition by *Tg*ApiAT6-1 and *Tg*ApiAT1

In order to understand the integrated contributions of both *Tg*ApiAT6-1 and *Tg*ApiAT1 to parasite AA$^+$ acquisition, we investigated the transport mechanisms of the two proteins in more detail. Our observation of *Tg*ApiAT6-1-mediated Arg outward currents (Fig 3B and 3C), and our previous work on the related aromatic and neutral amino acid exchanger *Tg*ApiAT5-3 [20], suggested that *Tg*ApAT6-1 and *Tg*ApiAT1 might operate as (bidirectional) amino acid exchangers. To explore this possibility, we first asked whether *Tg*ApiAT6-1 and *Tg*ApiAT1 could efflux substrates. Oocytes expressing *Tg*ApiAT6-1 were preloaded with [14C]Lys and the efflux of radiolabel was measured in the presence or absence of 1 mM unlabelled Lys in the extracellular medium (Fig 5A). Similarly, oocytes expressing *Tg*ApiAT1 were preloaded with [14C]Arg and the efflux of radiolabel was measured in the presence or absence of 1 mM unlabelled extracellular Arg (Fig 5B). For both *Tg*ApiAT6-1 and *Tg*ApiAT1, the presence of substrate in the extracellular medium gave rise to an *increase* in effluxed [14C]substrate and a *decrease* in oocyte-retained [14C]substrate over time (Fig 5A and 5B); *i.e.* extracellular Lys induced a '*trans*-stimulation' of the efflux of [14C]Lys from oocytes expressing *Tg*ApiAT6-1, and extracellular Arg induced a *trans*-stimulation of the efflux of [14C]Arg from oocytes expressing *Tg*ApiAT1. Neither efflux of preloaded substrate nor *trans*-stimulation of either substrate was observed in H₂O-injected control oocytes (S8A and S8B Fig). These results indicate that, in addition to the net substrate uptake that we have demonstrated previously (Fig 2; [27]), both transporters are capable of substrate exchange and can be *trans*-stimulated; *i.e.* the unidirectional flux of radiolabel is stimulated by the presence of substrate at the opposite (*trans*) face of the membrane.

Our earlier results (Fig 2) indicated that *Tg*ApiAT6-1 could also mediate the flux of some neutral amino acids. To examine whether other substrates *trans*-stimulate AA$^+$ flux via *Tg*ApiAT6-1 and *Tg*ApiAT1, we measured the efflux of [14C]Arg via *Tg*ApiAT6-1 and *Tg*ApiAT1 using a range of amino acids and amino acid derivatives as counter-substrates. The strongest *trans*-simulation of *Tg*ApiAT6-1-mediated [14C]Arg efflux was observed in the presence of several cationic amino acids (*e.g.* Arg, His, Orn), as well as by the large neutral amino acids Leu and Met (Fig 5C). Notably, when Lys was used as a counter-substrate for *Tg*ApiAT6-1, Arg efflux was significantly *lower* than when measured in the absence of a counter-substrate. As the rate of transport for any substrate is determined by the slowest step in the transport mechanism (*i.e.* by the rate-limiting step of the transport cycle), we hypothesise that the counter-transport of extracellular Lys via *Tg*ApiAT6-1 into the oocyte represents a rate-limiting step in which the rate at which [14C]Arg efflux can occur via *Tg*ApiAT6-1 is limited by the rate at which (unlabelled) Lys is transported into the oocyte. This notion is supported by the very low maximal Lys transport rate relative to maximal Arg transport rate (see Fig 2E and 2F

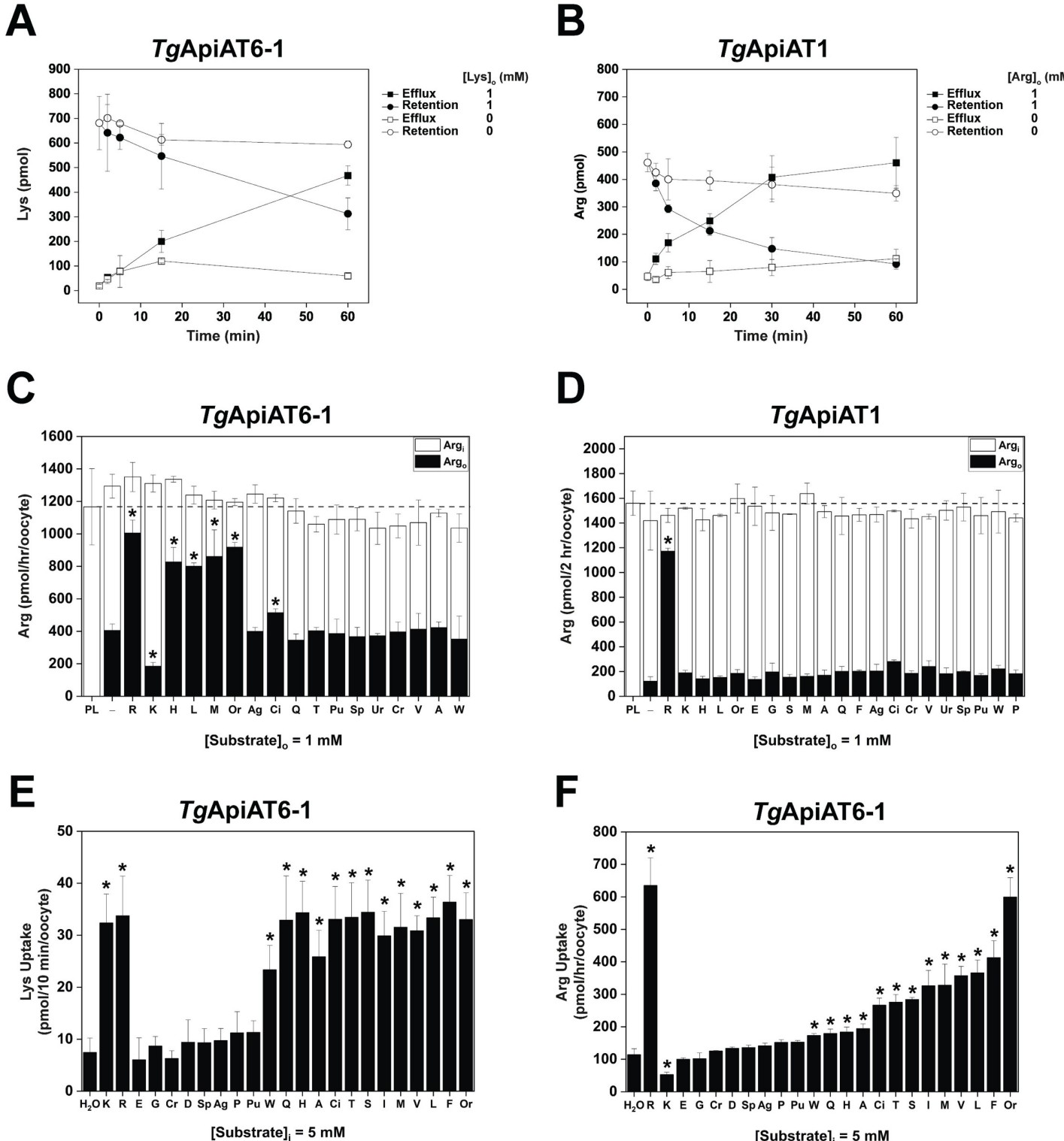

**Fig 5. *Trans*-simulation of AA+ transport in *Tg*ApiAT6-1- and *Tg*ApiAT1-expressing oocytes. A-B.** *Tg*ApiAT6-1-injected (A) and *Tg*ApiAT1-injected (B) oocytes were pre-loaded with either 1 mM unlabelled Lys and 1.0 μCi/ml of [14C]Lys (*Tg*ApiAT6-1) or 1 mM unlabelled Arg and 1.0 μCi/ml of [14C]Arg (*Tg*ApiAT1) for 3–6 hr. The retention of substrates in *Tg*ApiAT6-1- or *Tg*ApiAT1-expressing oocytes were measured in the presence of 1 mM external substrate (closed symbols) or in the absence of an external substrate (open symbols). Data points represent the mean ± S.D. from 3 batches of 5 oocytes from one experiment, and are representative of 3 independent experiments. **C-D.** Arg efflux and retention in *Tg*ApiAT6-1 expressing (C) and *Tg*ApiAT1 expressing (D) oocytes in the presence of candidate *trans*-stimulating substrates. Oocytes were pre-loaded (PL) with Arg by first microinjecting Arg to a final concentration of ~5 mM, then incubated oocytes in a solution

containing 1 mM unlabelled Arg and 1.0 µCi/ml of [14C]Arg for 1 hr (*Tg*ApiAT6-1) or 2 hr (*Tg*ApiAT1). This pre-loading was followed by addition of 1 mM unlabelled amino acids or amino acid derivatives to the outside of the oocyte. Efflux of pre-loaded Arg (Arg$_o$; black bars) and retention of pre-loaded Arg (Arg$_i$; white bars) were measured after 1 hr (*Tg*ApiAT6-1) or 2 hr (*Tg*ApiAT1) in the presence of 1 mM of the metabolites or with ND96 buffer in the extracellular medium (−). The horizontal dashed line across both figures indicates the amount of Arg pre-loaded (PL) into oocytes (left-most bar). Amino acid substrates are represented by single letter codes, while for other metabolites: Cr, creatine; Ag, agmatine; Sp, spermidine; Pu, putrescine; Ci, citrulline; Ur, urea; and Or, ornithine. Each bar represents the mean ± S.D. efflux or retention in from 3 batches of 5 oocytes from one experiment, and are representative of three independent experiments. Statistical analysis compares all bars to oocyte efflux in the presence of ND96 (−) (pH 7.3) (* $P < 0.05$, one-way ANOVA, Dunnett's post-hoc test). **E-F**. Lys (E) or Arg (F) uptake into *Tg*ApiAT6-1 expressing oocytes pre-loaded with a range of candidate *trans*-stimulating substrates. *Tg*ApiAT6-1-injected oocytes were pre-loaded by microinjecting the indicated substrates to a final concentration of ~5 mM (or with the same volume of H$_2$O as a control), and the uptake of 15 µM Lys and 1.0 µCi/ml of [14C]Lys (E) or 1 mM Arg and 1.0 µCi/ml of [14C]Arg (F) was determined. Uptake of Arg and Lys into control oocytes not expressing *Tg*ApiAT6-1 using the same *trans*-stimulation conditions (shown in S3D and S3E Fig for Arg and Lys uptake, respectively) were subtracted for all conditions. Amino acid substrates are represented by single letter codes, while for other metabolites: Cr, creatine; Ag, agmatine; Sp, spermidine; Pu, putrescine; Ci, citrulline; and Or, ornithine. Each bar represents the mean ± S.D. uptake of 10 oocytes for a single experiment, and are representative of three independent experiments. Statistical analysis compares all bars to Lys or Arg uptake in control H$_2$O 'pre-loaded' oocytes (* $P < 0.05$, one-way ANOVA, Dunnett's post-hoc test).

and Table 1). We observed significant *trans*-stimulation of Lys efflux by large neutral amino acids and by cationic amino acids, although, unlike the effects of Lys on Arg efflux, none of the tested counter substrates inhibited Lys efflux (S3F Fig).

In contrast to the range of cationic and neutral amino acids that were able to *trans*-stimulate Arg efflux by TgApiAT6-1, we were unable to detect *trans*-stimulation of [14C]Arg efflux through *Tg*ApiAT1 using any counter-substrate other than Arg itself (Fig 5D). This suggests that *Tg*ApiAT1 does not transport substrates other than Arg, and is consistent with our previous study that indicated *Tg*ApiAT1 is a highly-selective Arg transporter [27].

To determine whether the specificity of *trans*-stimulation holds true for transport in both directions, we reversed the direction of substrate flux in *Tg*ApiAT6-1 expressing oocytes, and measured the *trans*-stimulation of Lys *uptake* by a range of substrates. Oocytes expressing *Tg*ApiAT6-1 were microinjected with a range of amino acids and arginine-derived metabolites to a final concentration of approximately 5 mM and the uptake of [14C]Lys was measured. Cationic amino acids and a number of neutral and hydrophilic amino acids *trans*-stimulated Lys uptake via *Tg*ApiAT6-1 (Fig 5E). None of the *trans*-stimulating amino acids increased the rate of Lys uptake beyond that observed under conditions of *trans*-stimulation by intracellular Lys. Next, we measured *trans*-stimulation of the uptake of [14C]Arg via *Tg*ApiAT6-1. As observed with the efflux experiments, several cationic (Arg, Orn) and large neutral amino acids (Val, Leu, Met, Phe) *trans*-stimulated Arg uptake into *Tg*ApiAT6-1-expressing oocytes (Fig 5F). By contrast, uptake of Arg with Lys present on the other side of the membrane was lower than for any other *trans*-stimulating substrate, and lower even than non-*trans*-stimulated uptake. This mirrors our observation of reduced Arg efflux when external Lys is present (Fig 5C), and further supports the hypothesis that the slow counter-transport of Lys acts as a rate-limiting step in the transport cycle of *Tg*ApiAT6-1 under the conditions of these transport assays.

In summary, these experiments indicate that transport of AA$^+$ by *Tg*ApiAT6-1 can be *trans*-stimulated by a range of cationic and neutral amino acids at both the intra- and extracellular face of the membrane. Together these results are consistent with Lys being a high-affinity but low V$_{max}$ substrate of *Tg*ApiAT6-1 in comparison to Arg, which has a lower affinity for the transporter but a much higher maximal rate of transport. The data in Fig 5C and 5F are also consistent with the low maximal rate of Lys transport by *Tg*ApiAT6-1 setting an upper limit (rate-limitation) to the speed at which Arg can be taken up or effluxed by *Tg*ApiAT6-1 under conditions in which Lys is present.

Our data indicate that *Tg*ApiAT1, a highly selective Arg transporter, is *trans*-stimulated strongly by Arg (Fig 5D). This could limit the net accumulation of Arg within parasites, with one molecule of Arg effluxed for every molecule that is transported in. Similarly, *Tg*ApiAT6-1, which exhibits little unidirectional efflux in the absence of *trans*-substrate and has a higher

affinity for Lys than other amino acids, may be limited in its capacity to accumulate Lys and other substrates. We therefore utilised the oocyte expression system to investigate whether *Tg*ApiAT6-1 and *Tg*ApiAT1 are capable of net substrate accumulation, testing whether the intracellular concentration of amino acid substrates reached a level higher than the extracellular concentration.

To measure whether AA$^+$ accumulation occurs in *Tg*ApiAT6-1 and *Tg*ApiAT1-expressing oocytes, we used a targeted LC-MS/MS approach to measure intra-oocyte substrate concentrations while accounting for the endogenous metabolism of these substrates [36]. *Tg*ApiAT6-1 or *Tg*ApiAT1-expressing oocytes were incubated in a solution containing 1 mM Lys or Arg for 50–54 hr, a timeframe that enabled us to observe the accumulation of specific amino acids over time, which we then quantified using external calibration curves (S9A and S9B Fig). *Tg*ApiAT6-1 expressing oocytes accumulated Lys to an intracellular concentration more than two-fold higher than the extracellular concentration, with full electrochemical equilibrium not yet reached at the final time point (Fig 6A, closed squares). Similarly, Arg accumulated to an estimated intracellular concentration some four- to five-fold higher than the extracellular concentration in oocytes expressing *Tg*ApiAT6-1, reaching equilibrium after ~40 hr (Fig 6B, closed squares). By contrast, in H$_2$O-injected control oocytes the intracellular concentrations of Lys and Arg increased to a value that was approximately the same as the extracellular concentration after ~24 hr, remaining at this level for the duration of the experiment (Fig 6A and 6B, closed circles). Of the metabolites reliably identified by LC-MS/MS across all conditions, only Lys and metabolic products of Lys (S1 Table) and Arg (S2 Table) displayed large intracellular accumulation in AA$^+$-incubated oocytes expressing *Tg*ApiAT6-1.

Removing the extracellular substrate from *Tg*ApiAT6-1 expressing oocytes after 32–34 hr resulted in a decrease in intracellular concentrations of both Lys and Arg (Fig 6A and 6B, open squares); this was not observed in H$_2$O-injected oocytes (open circles) and therefore is unlikely to be a result of these amino acids being incorporated into proteins and/or by conversion into other metabolites. Instead, these data are consistent with *Tg*ApiAT6-1 mediating the net efflux of amino acids from oocytes when external substrate is absent. Together with other results, these data indicate that *Tg*ApiAT6-1 is able to mediate the accumulation of cationic amino acids.

*Tg*ApiAT1 also mediated a substantial accumulation of Arg, with the intracellular concentration of Arg reaching a level some three-fold higher than the extracellular concentration after 32 hr (Fig 6C, closed squares), then decreasing following the removal of Arg from the medium (Fig 6C, open squares). Oocytes expressing *Tg*ApiAT1 displayed a slower accumulation of Arg than did oocytes expressing *Tg*ApiAT6-1. As was observed for oocytes expressing *Tg*ApiAT6-1, Arg was the only compound shown to undergo substantial intracellular accumulation in oocytes expressing *Tg*ApiAT1 and incubated in the presence of extracellular Arg (S2 Table).

Together, these data indicate that *Tg*ApiAT6-1 and *Tg*ApiAT1 function as uniporters, with the capacity to mediate amino acid exchange. Both transporters have the capacity to accumulate cationic substrate to concentrations higher than that in the extracellular medium. However, *Tg*ApiAT6-1 has a broad specificity for many AA$^+$, large neutral amino acids and arginine metabolites and a very high selectivity for Lys, such that when Lys is present it reduces the transport rate for all other substrates. By contrast, Arg transport by *Tg*ApiAT1 is *trans*-stimulated solely by Arg (and no other amino acid), consistent with previous data indicating the high selectivity of this transporter for Arg [27].

## Discussion

Our study establishes that *Tg*ApiAT6-1 is essential for tachyzoite proliferation *in vitro*, most likely due to its role in uptake of the essential amino acid Lys. However, *Tg*ApiAT6-1 may also

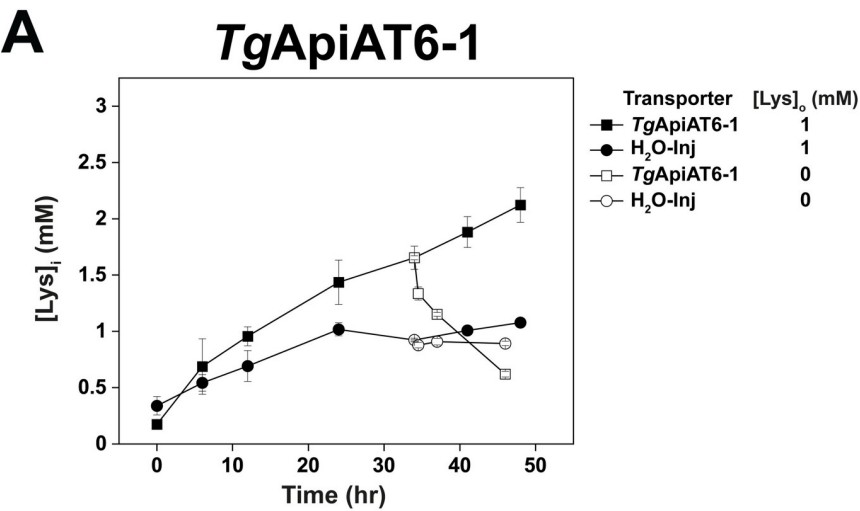

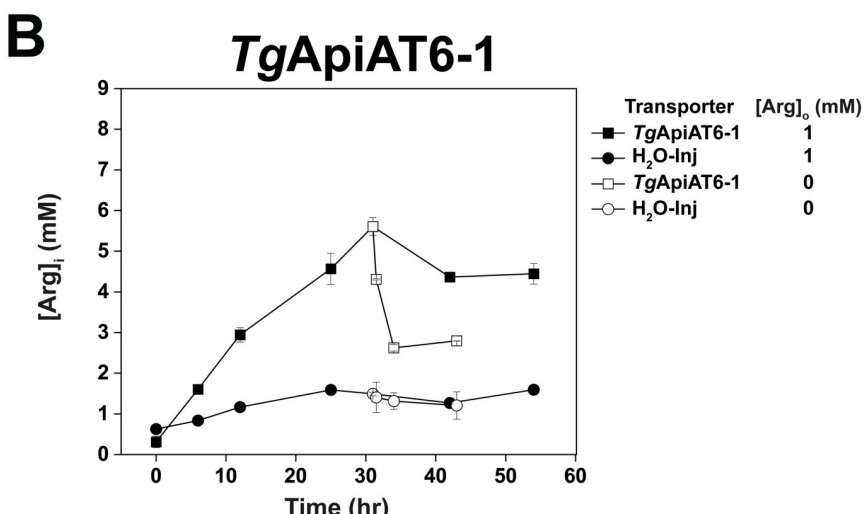

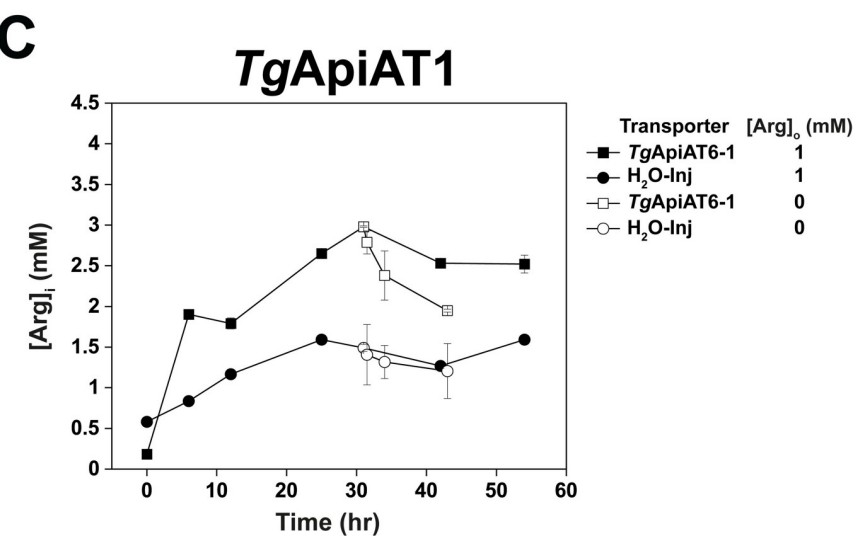

**Fig 6. *Tg*ApiAT6-1 and *Tg*ApiAT1 are net accumulators of cationic amino acids. A-C.** Time-course measuring the Lys (A) or Arg (B-C) concentration in *Tg*ApiAT6-1 expressing oocytes (A-B; squares), *Tg*ApiAT1 expressing oocytes (C; squares), or H₂O-injected oocytes (A-C; circles) incubated in 1 mM Lys (A) or 1 mM Arg (B-C) as quantified by LC-MS/MS. Following 32–34 hr of incubation measuring accumulation inside oocytes, samples were split into two groups, one continuing with substrate incubation (closed symbols), the other with substrate replaced by substrate-free incubation media (open symbols). Each data point represents the mean intracellular amino acid concentration ± S.D. of 12 individual oocytes (substrate-incubated) or 3 batches containing 5 oocytes each (substrate-free), and are representative of 3 independent experiments.

contribute to the uptake of other cationic and neutral amino acids and amino acid derivatives, particularly Arg, *in vivo*. In a previous study, we identified *Tg*ApiAT1 as a selective Arg transporter, and predicted the existence of a broad AA⁺ transporter that serves as an alternative Arg uptake pathway to *Tg*ApiAT1 [27]. The data in the present study indicate that *Tg*ApiAT6-1 functions as this alternative Arg transporter. We have recently shown that the expression of *Tg*ApiAT1 is up-regulated under Arg-limiting conditions and expressed at low levels under Arg-replete conditions [31]. The differential expression of *Tg*ApiAT1 may therefore allow these parasites to survive when Arg levels are limited, while *Tg*ApiAT6-1 may ensure regulated uptake of Arg and Lys under nutrient-rich conditions.

A recent study demonstrated that intracellular *T. gondii* tachyzoites activate an integrated stress response pathway in host cells, leading to an increase in the abundance of the mammalian cationic amino acid transporter CAT1, and a subsequent increase in Arg uptake into host cells [41]. Like *Tg*ApiAT6-1, CAT1 is capable of both Lys and Arg uptake. However, in contrast to *Tg*ApiAT6-1, CAT1 has a similar affinity for the two amino acids [42]. Upregulation of CAT1 is therefore likely to also cause an increase in Lys uptake in host cells in which CAT1 serves as the major AA⁺ transporter. We have shown previously that the ratio of [Arg]:[Lys] in the extracellular medium (rather than the absolute concentrations of each) determines the importance of *Tg*ApiAT1 for parasite proliferation [27]. Our finding that *Tg*ApiAT6-1 is a high affinity Lys transporter provides an explanation for this observation: Arg uptake by *Tg*ApiAT6-1 is only possible when [Arg] is high enough for Arg to out-compete Lys for uptake by this transporter, or when Lys levels are low. It remains to be seen whether Lys and Arg uptake into host cells increases in response to parasite infection in host organs in which other host AA⁺ transporters contribute to uptake (e.g. in the liver, where CAT2 has a major role in facilitating AA⁺ uptake; [43]). In this context, it is notable that liver stage development of *P. berghei* parasites is impaired in mice in which CAT2 has been knocked out [44], and it is conceivable that the roles of *Tg*ApiAT6-1 and *Tg*ApiAT1 in Arg uptake may differ in host cells that express different AA⁺ transporters.

Based on our findings, and on several other recent studies into Arg uptake in *T. gondii* [27,31,41], we can now propose a comprehensive model for the uptake of cationic amino acids into these parasites (Fig 7). The scavenging of AA⁺ by parasites results in a depletion of these amino acids in the host cell cytosol, causing upregulation of the host CAT1 AA⁺ transporter [41]. This increases AA⁺ uptake into host cells in which CAT1 is the major AA⁺ transporter, while maintaining the Arg:Lys ratio from the extracellular milieu. Lys is a high affinity substrate for *Tg*ApiAT6-1, and is taken up into parasites through this transporter in all intracellular niches (Fig 7). If the ratio of Arg:Lys in the host cell is low (*e.g.* in the cells of organs with high Arg catabolism such as the liver [45]; Fig 7, right), Lys uptake by *Tg*ApiAT6-1 will outcompete Arg uptake. The parasite responds by upregulating *Tg*ApiAT1, which enables sufficient Arg uptake for parasite proliferation [31]. If the ratio of Arg:Lys in the host cell is high (*e.g.* in the cells of organs like the kidneys that function in the net synthesis of Arg [46]; Fig 7, left), Arg can compete with Lys for uptake by *Tg*ApiAT6-1, resulting in an increased role for

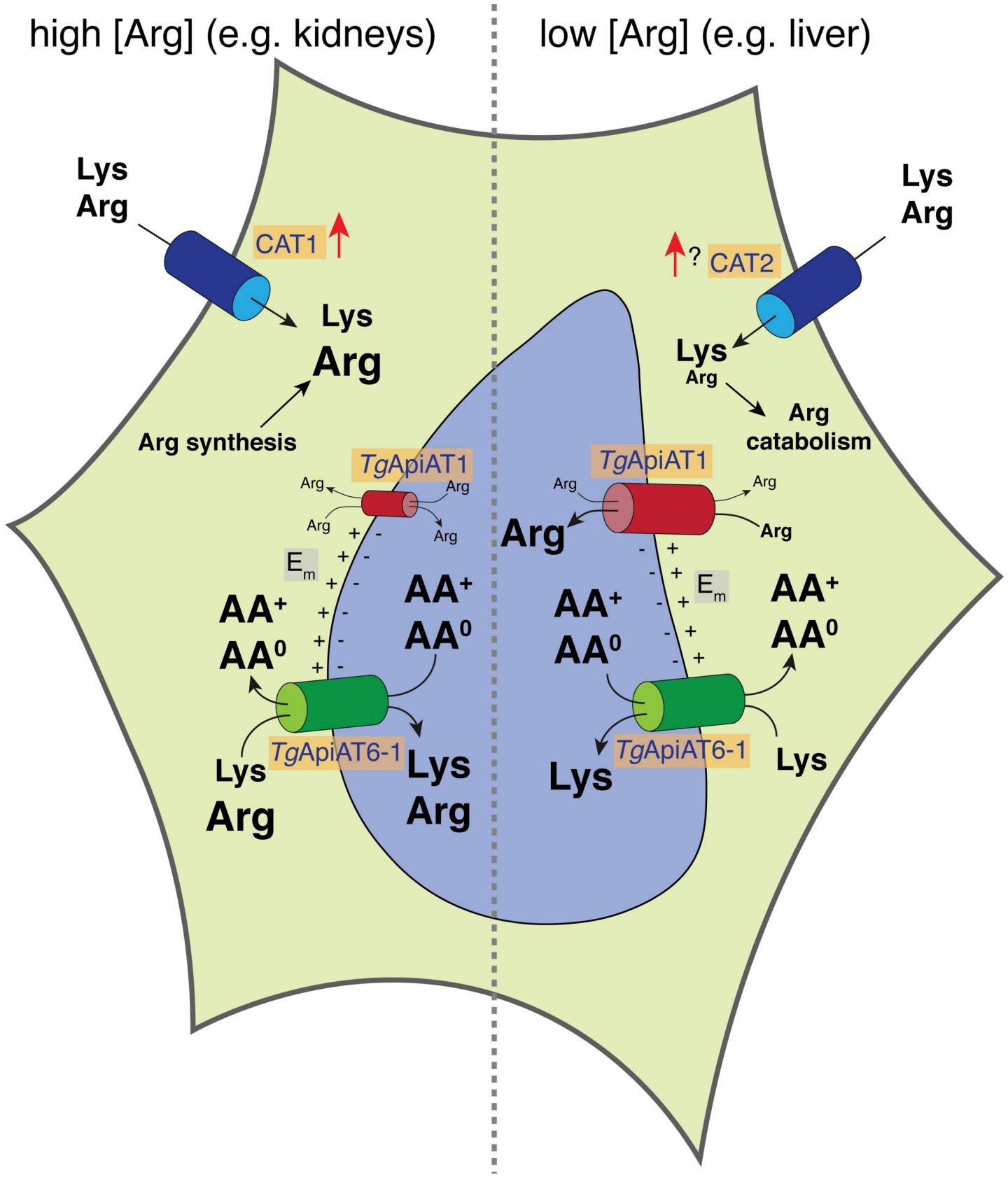

**Fig 7. Model for AA$^+$ uptake into intracellular *T. gondii* parasites.** The proliferation of *T. gondii* parasites (light blue) inside infected host cells (yellow) causes a depletion of host cell Arg, leading to an upregulation of the host cell AA$^+$ transporters CAT1 and, possibly, CAT2 (dark blue), and a concomitant increase in the uptake of Lys and Arg into host cells. In organs with high Arg catabolism (*e.g.* liver, right), the intracellular ratio of [Arg]:[Lys] is low, and Lys uptake through *Tg*ApiAT6-1 (green) outcompetes Arg uptake through this transporter. The parasite responds by upregulating the abundance of its selective Arg transporter, *Tg*ApiAT1 (red), enabling Arg uptake through this transporter. In organs in which Arg is synthesised (*e.g.* kidneys, left), the intracellular ratio of [Arg]:[Lys] is high, resulting in increased uptake of Arg through *Tg*ApiAT6-1. Parasites respond by downregulating *Tg*ApiAT1 abundance. The transport activity of *Tg*ApiAT6-1 is increased by the exchange of Lys and Arg with cationic (AA$^+$) and neutral (AA$^0$) amino acids. The activity of both *Tg*ApiAT1 and *Tg*ApiAT6-1 may be increased by an inwardly negative membrane potential (E$_m$) at the parasite plasma membrane.

*Tg*ApiAT6-1 in Arg uptake in such environments (and a decreased role for *Tg*ApiAT1, corresponding to its decreased abundance; [31]).

Together, the ability of *Tg*ApiAT6-1 and *Tg*ApiAT1 to efflux substrate without the presence of *trans* substrate (Fig 6), the net Arg-carried inward currents observed using electrophysiological recordings (Fig 3), and the *trans*-stimulation experiments (Fig 5), indicate that both transporters are uniporters with the capacity to mediate substrate exchange. We demonstrate that both *Tg*ApiAT6-1 and *Tg*ApiAT1 have the capacity to accumulate substrates to a concentration higher than the extracellular concentration when expressed in oocytes (Fig 6), and we propose that the same holds true in parasites. One way a cationic substrate could be favoured for accumulation via net uptake is to harness the negative inside membrane potential (E$_m$) that is present across the plasma membrane of many cells (including extracellular *T. gondii* parasites; Fig 7; [47]), which would naturally favour accumulation of cationic substrates. Theoretically, at the resting E$_m$ values for oocytes, which we determined to be between −30 and −40 mV (S9C Fig), AA$^+$ could accumulate from between 3.2- to 4.7-fold higher intracellular concentration than extracellular concentration for *Tg*ApiAT6-1 or *Tg*ApiAT1 expressing oocytes (S9D Fig). This predicted accumulation is consistent with the observed four- to five-fold accumulation of Arg by oocytes expressing *Tg*ApiAT6-1.

As well as influencing the equilibrium of substrate, the inwardly negative E$_m$ may also function to change the activation energy and, therefore, the probability of carrier-substrate versus carrier-free conformational cycling across the membrane in both transporters. Alternatively, the inward negative E$_m$ could reduce the affinity for binding of AA$^+$ when the transporter is inward facing. As the *Tg*ApiAT6-1-mediated currents are the net real-time inward translocation of charge, they represent the balance between inward and outward transport and hence a read-out of carrier-substrate versus carrier-free movements. The effect of E$_m$ on inward directed substrate affinity is supported by the considerable differences in K$_{0.5}$ values calculated for Arg transport by *Tg*ApiAT6-1 between the [$^{14}$C]Arg uptake and electrophysiology experiments. In the [$^{14}$C]Arg uptake experiments, in which the resting E$_m$ of oocytes is between −30 to −40 mV (S9C Fig), we determined the K$_{0.5}$ to be 748 μM (Table 1), higher than the 190 μM K$_{0.5}$ value calculated in the Arg-dependent current recordings, which were conducted under conditions in which the voltage was clamped to −50 mV (Fig 3E). An increased inwardly negative E$_m$, therefore, correlates with increased affinity of *Tg*ApiAT6-1 for Arg.

Extracellular *T. gondii* tachyzoites are known to have an inwardly negative E$_m$ [47]. Whether the same is true of intracellular tachyzoites, the stage at which parasite proliferation is dependent on Arg and Lys uptake, has not been determined. We note, however, that many other organisms, including *P. falciparum*, utilise the membrane potential to energise the import and efflux of desired substrates (*e.g.* [48,49]). Alternatively, it is plausible that an inwardly negative E$_m$ is not absolutely necessary for net substrate accumulation, as the metabolism of both amino acids in processes such as protein synthesis (with a concomitant decrease in their intracellular pools) would also drive uptake.

We demonstrate that AA$^+$ uptake via *Tg*ApiAT6-1 can be *trans*-stimulated by AA$^+$ and large neutral amino acids inside the cell, increasing the rate of AA$^+$ accumulation into parasites

(Fig 7). This may be beneficial in an environment in which parasites are competing with their host cells for these essential nutrients and may also involve the coordinated action of both *Tg*ApiAT1 and *Tg*ApiAT6-1, with accumulation of Arg by the former facilitating the faster accumulation of Lys by the latter. In human cells, the coordinated action of different amino acid transporters has been shown to play a role in the efficient uptake of essential amino acids by both normal and malignant cells [50,51], and we propose the same is true for AA$^+$ uptake in *T. gondii.*

The specificity of a transporter with high exchange capacity will affect its ability to accumulate selective substrates [52]. For example, if a transporter has higher affinity for one substrate over others–as *Tg*ApiAT6-1 has for Lys over Arg and neutral amino acids–a broad specificity would assist in the preferred accumulation of the high affinity substrate. Under conditions of low intracellular concentrations (high metabolic demand) of the preferred substrate (Lys in the case of *Tg*ApiAT6-1), the presence of high levels of lower affinity substrates in the cytosol (Arg or neutral amino acids in the case of *Tg*ApiT6-1), would facilitate the rapid *trans*-stimulated uptake of the preferred substrate (Fig 7; [53]).

In summary, the transport mechanism of *Tg*ApiAT6-1, elucidated in this study, is well-adapted for enabling the coordinated acquisition of essential cationic amino acids by the *T. gondii* parasite. The faster overall uptake rate and much higher $V_{max}$ for Arg compared to Lys for *Tg*ApiAT6-1 means that this transporter is able to meet the residual demand for Arg uptake in Arg-replete conditions. We propose that the energisation of the uptake of AA$^+$ by the parasite's inwardly negative $E_m$, together with the *trans*-stimulation of uptake by excess AA$^+$ and large neutral amino acids, facilitates the net accumulation of both Lys and Arg via this transporter.

Our study establishes the key role of *Tg*ApiAT6-1 in Lys and Arg uptake in *T. gondii* parasites, and our functional studies on the transport mechanisms of *Tg*ApiAT6-1 and *Tg*ApiAT1 reveal the complex and coordinated processes by which the uptake of these essential amino acids is mediated in a model intracellular parasite. The coordinated functioning of these transporters is likely to have contributed to the remarkably broad host cell range of these parasites.

## Materials and methods

### Ethics statement

*Xenopus laevis* frog maintenance and oocytes preparation was approved by the Australian National University Animal Experimentation Ethics Committee (Protocols A2014/20 and A2020/48). Before surgeries to extract oocytes, frogs were anesthetized by submersion in a 0.2% (w/v) tricaine methanesulfonate (MS-222) solution made in tap water and neutralized with $Na_2HCO_3$ for 15–40 minutes until frogs exhibited no reaction upon being turned upside down.

### Parasite culture and proliferation assays

Parasites were maintained in human foreskin fibroblasts in a humidified 37˚C incubator with 5% $CO_2$. Parasites were cultured in Roswell Park Memorial Institute 1640 (RPMI) medium supplemented with 1% (v/v) fetal calf serum, 2 mM L-glutamine, 50 U/ml penicillin, 50 µg/ml streptomycin, 10 µg/ml gentamicin, and 0.25 µg/ml amphotericin b. Where applicable, anhydrotetracycline (ATc) was added to a final concentration of 0.5 µg/ml with 0.025% (v/v) ethanol added to relevant vehicle controls. In experiments wherein we varied the concentration of Lys in the growth medium, we generated homemade RPMI medium containing a range of [Lys], as described previously [31].

Fluorescence proliferation assays were performed as described previously [27]. Briefly, 2,000 tdTomato-expressing parasites were added to wells of an optical bottom 96-well plate containing a monolayer of host cells. Fluorescence was read regularly using a FLUOstar Optima plate reader (BMG).

## Generation of genetically modified *T. gondii* parasites

To generate a *T. gondii* strain in which we could knock down expression of *Tg*ApiAT6-1, we replaced the native *Tg*ApiAT6-1 promoter region with an ATc-regulatable promoter using a double homologous recombination approach. First, we amplified the 5' flank of *Tg*ApiAT6-1 with the primers ApiAT6-1 5' flank fwd (5'-GACTGGGCCCCTTCATTTCTTCGCAACGTG ACAAGC) and ApiAT6-1 5' flank rvs (5'-GACTCATATGCCGACTTGCTTGAAGAACCT GCG). We digested the resulting product with *Psp*OMI and *Nde*I and ligated into the equivalent sites of the vector pPR2-HA₃ [54], generating a vector we termed pPR2-HA₃(ApiAT6-1 5' flank). Next, we amplified the 3' flank with the primers ApiAT6-1 3' flank fwd (5'- GATCA-GATCTAAAATGGCGTCCTCGGACTCGAAC) and ApiAT6-1 3' flank rvs (5'-CTAGG CGGCCGCGAGTTCGGAGGACGATCCAGAGG). We digested the resulting product with *Bgl*II and *Not*I and ligated into the equivalent sites of the vector pPR2-HA₃(ApiAT6-1 5' flank) vector. We linearized the resulting plasmid with *Not*I and transfected this into TATi/Δ*ku80* strain parasites [55] expressing tandem dimeric Tomato RFP [27]. We selected parasites on pyrimethamine and cloned parasites by limiting dilution. We screened clones for successful integration of the ATc regulatable promoter using several combinations of primers. To test for 3' integration, we used the ApiAT6-1 3' screen fwd (5'-CCGCAGTGGACGGACACC) and ApiAT6-1 3' screen rvs (5'-CAGTTCCGCTCGGTTGCTTG) primers to detect the presence of the native locus, and the t7s4 screen fwd (5'- ACGCAGTTCTCGGAAGACG) and ApiAT6-1 3' screen rvs (above) primers to detect the presence of the modified locus. To screen for 5' integration, we used the ApiAT6-1 5' screen fwd (5'-CTGGAGAAGTGTGTGAGGAGC) and ApiAT6-1 screen rvs (5'-GAGTGGAGACGCTGCGACG) primers to test for the presence of the native locus, and ApiAT6-1 5' screen fwd (above) and DHFR screen rvs (5'-GGTGTCGT GGATTTACCAGTCAT) primers to detect the presence of the modified locus. We termed the resulting strain regulatable (r)*Tg*ApiAT6-1. To enable us to measure knockdown of the ATc-regulatable *Tg*ApiAT6-1 protein, we integrated a HA tag into the r*Tg*ApiAT6-1 locus by transfecting a *Tg*ApiAT6-1-HA 3' replacement vector, described previously [20], into this strain.

To complement the r*Tg*ApiAT6-1 strain with a constitutively-expressed copy of *Tg*ApiAT6-1, we amplified the *Tg*ApiAT6-1 open reading frame with the primers *Tg*ApiAT6-1 orf fwd (5'-GATCAGATCTAAAATGGCGTCCTCGGACTCGAAC) and *Tg*ApiAT6-1 orf rvs (5'- GATCCCTAGGAGCGGAGTCTTGCGGTGGC) using genomic DNA as template. We digested the resulting product with *Bgl*II and *Avr*II and ligated into the equivalent sites of the pUgCTH₃ vector [27]. We linearised the resulting vector with *Mfe*I, transfected into r*Tg*ApiAT6-1 parasites, and selected on chloramphenicol. We cloned drug resistant parasites before subsequent characterisation.

To produce a parasite strain containing a frameshift mutation in *Tg*LysA, we first generated a single guide RNA (sgRNA)-expressing vector that targeted the *Tg*LysA locus. To do this, we introduced a sgRNA-encoding sequence targeting *Tg*LysA into the pSAG1::Cas9-U6::sgUPRT vector (Addgene plasmid #54467) by Q5 mutagenesis, as described previously [56], using the primers *Tg*LysA CRISPR fwd (5'-AGGCGCCTCCAAATTTCTCAGTTTTAGAGCTAGAAA-TAGCAAG) and universal rvs (5'-AACTTGACATCCCCATTTAC). We transfected the resulting vector into RHΔ*hxgprt*/Tomato parasites [57], then cloned Cas9-GFP-expressing parasites by fluorescence activated cell sorting 3 days after transfection. We amplified and

sequenced the region around the *Tg*LysA mutation site in clonal parasites with the primers *Tg*LysA seq fwd (5'-GCTTGGCTGAGCTTTTGCT) and *Tg*LysA seq rvs (5'-GCACTG-CAGGTTGACTTGG). We selected a clone containing a 2 bp deletion in the locus for subsequent characterisation.

## Radiolabel uptake assays in *T. gondii* parasites

Radiolabel uptake assays with extracellular *T. gondii* parasites were performed as described previously [58]. Briefly, WT and r*Tg*ApiAT6-1 parasites were cultured in Dulbecco's Modified Eagle's medium (containing 400 μM Arg and 800 μM Lys) in the absence or presence of ATc for 2 days. Parasites were isolated from host cells and resuspended in Dulbecco's PBS (pH 7.4) supplemented with 10 mM glucose (PBS-glucose) and a mix of radiolabelled and unlabelled amino acids. Specifically, Lys uptake was measured by incubation in 0.1 μCi/ml [$^{14}$C]Lys and 50 μM Lys, Arg uptake was measured by incubation in 0.1 μCi/ml [$^{14}$C]Arg and 80 μM unlabelled Arg, and 2-deoxyglucose (2-DOG) was measured by incubation in 0.2 μCl/ml [1-$^{14}$C] 2-DOG and 25 μM unlabelled 2-DOG. Samples were incubated at 37˚C, with aliquots removed at regular intervals and centrifuged through an oil mixture containing 84% (v/v) PM-125 silicone fluid and 16% (v/v) light mineral oil to separate parasites from unincorporated radiolabel. Samples were lysed and incorporated radiolabel was measured using a scintillation counter.

## Parasite protein detection

Parasite protein preparations and associated western blotting was performed as described previously [59]. Membranes were probed with rat anti-HA (Sigma, clone 3F10; 1:500 dilution) and rabbit anti-*Tg*Tom40 ([60]; 1:2,000 dilution) primary antibodies, and horseradish peroxidase-conjugated goat anti-rat (Abcam catalogue number ab97057; 1:5,000 dilution) and goat anti-rabbit (Abcam catalogue number ab97051; 1:10,000 dilution) secondary antibodies. Blots were imaged on a ChemiDoc MP imaging system (Biorad).

## [$^{13}$C]-Amino acid labelling in parasites

Labelling experiments with a mixture of [$^{13}$C]-amino acids were performed as described previously [20]. Briefly, r*Tg*ApiAT6-1 parasites were cultured for 2 days in the absence or presence of ATc. Egressed parasites were incubated in amino acid-free RPMI supplemented with 2 mg/ml algal [$^{13}$C]amino acid mix (Cambridge Isotope Laboratories, catalogue number CLM-1548) for 15 min at 37˚C, and parasite metabolites extracted in chloroform:methanol:water (1:3:1 v/v/v) containing 1 nmol *scyllo*-inositol as an internal standard. Polar metabolites in the aqueous phase were dried, methoxyamated (20 mg/ml methoxyamine in pyridine overnight), then trimethylsilylated (N,O-bis(trimethylsilyl)trifluoroacetamide with 1% trimethylsilyl) for 1 hr at room temperature. Samples were analysed using GC-MS as described [61], and the fractional labelling of detected amino acids was determined using DExSI software [62].

## *Xenopus laevis* oocytes preparation and transporter expression

*Xenopus laevis* oocytes were removed by abdominal surgical incision and extraction as previously reported [63]. To generate a vector from which we could express complementary RNA (cRNA) encoding the HA-tagged *Tg*ApiAT6-1 protein, we amplified the *Tg*ApiAT6-1 open reading frame with the primers *Tg*ApiAT6-1 oocyte fwd (5'-GATCCCCGGGCCAC-CATGGCGTCCTCGGACTCGAAC) and *Tg*ApiAT6-1 orf rvs (above). We digested the resulting product with *Xma*I and *Avr*II and ligated into the equivalent sites of the vector pGHJ-HA [27]. We prepared cRNA for injection and *in vitro* protein expression in oocytes as

previously described [63–65]. The pGHJ-*Tg*ApiAT6-1 and pGHJ-*Tg*ApiAT1 [27] plasmids were linearized by digestion with *Not*I (NEB) for 2 hr. Following *in vitro* synthesis and purification, cRNA was quantified using a Tecan Infinite M1000 Pro (Tecan Group, Männedorf, Switzerland) spectrophotometer ($OD_{260}$/$OD_{280}$). For all transporter assays in oocytes, cRNA was micro-injected into stage 5 or 6 oocytes using a Micro4 micro-syringe pump controller and A203XVY nanoliter injector (World Precision Instruments, Sarasota, FLA, U.S.A.). Oocytes were maintained in oocyte Ringer's buffer with added gentamycin (50 μg/ml) and $Ca^{2+}$ (1.8 mM) as previously described [66]. We determined the optimal expression conditions for *Tg*ApiAT6-1 such that all oocyte experiments were conducted on days 4 to 6 post-cRNA injection and by micro-injection of 15ng cRNA/oocyte (S2B and S2C Fig).

### *Xenopus laevis* oocyte uptake and efflux assays

Methods optimised for the study of ApiAT family transporters in *X. laevis* oocytes have been detailed previously [20]. For simple radiolabel uptake experiments, oocytes were washed four times in ND96 buffer (96 mM NaCl, 2 mM KCl, 1 mM $MgCl_2$, 1.8 mM $CaCl_2$, 5 mM HEPES, pH 7.3) at ambient temperature. In initial time-course experiments (S2D Fig) it was determined that the uptake of radiolabelled forms of both Lys and Arg were linear with time for over 10 min. In subsequent experiments estimates of initial uptake rates were therefore made using an incubation period of 10 mins except where indicated in the axis title. Uptake of radiolabel was quenched by washing oocyte batches four times in ice-cold ND96. For ion replacement uptake experiments in which alternative salt buffers were used, uptake was quenched by washing oocytes in the alternative buffer. The salt composition of these alternative buffers is indicated in the figure legends and detailed in S3 Table.

For uptake experiments in which *trans*-stimulation was investigated, 100 mM solutions of potential substrates (in ND96) were pre-injected at 25 nl/oocyte using a Micro4 micro-syringe and A203XVY nanoliter injector (World Precision Instruments). Oocytes were then incubated on ice for 30 mins prior to uptake experiments to allow for membrane recovery. The injection of 25 nl/oocyte of 100 mM solutions of the candidate substrates gave an estimated cytosolic concentration of 5 mM, based on an assumed free aqueous volume of 500 nl/oocyte [63,67]. Calculations of cytosolic concentrations from pre-injection should be treated as approximations only, as stage 5 or 6 oocyte diameters vary from 1–1.3 mm and aqueous oocyte volumes range from 368 to > 500 nl [67,68].

In conducting radiolabel efflux experiments, oocytes were pre-loaded with radiolabelled substrate using one of two different methods.

For the first method (Fig 5C and 5D), batches of 5 oocytes were pre-injected with unlabelled Arg calculated to an approximate cytosolic concentration of 5 mM as described in the preceding paragraph. This pre-injection was used to stimulate the subsequent loading (over a 1 or 2 hr incubation period) of 1 mM [$^{14}$C]Arg. After the loading period the extracellular radiolabel was washed away and efflux measurements were conducted.

For the second method (Figs 5A and 5B and S8A and S8B), radiolabelled substrate was simply added to the external oocyte-bathing solution and the samples were then incubated at 16–18˚C for 3–28 hr to allow the uptake of radiolabel. The loading-time varied, depending on whether oocytes were expressing *Tg*ApiAT6-1 or *Tg*ApiAT1 (in which case the radiolabel was taken up relatively quickly through these transporters) or whether the oocytes were the $H_2O$-injected controls (in which case radiolabel was taken up more slowly). The loading time was chosen so as to ensure that in each case the amount of radiolabel taken up by the oocytes was approximately the same. In the case of the $H_2O$-injected oocytes (*i.e.* oocytes not expressing *Tg*ApiAT6-1 or *Tg*ApiAT1) the preloading incubation time was typically 21 to 28 hr.

Pre-loading of oocytes with radiolabel was followed by quenching of the loading process by washing the oocytes in ice-cold ND96 solution, then initiation of the efflux by replacing the ice-cold solution with ambient-temperature solutions containing potential *trans*-stimulating substrates, as described in the figure legends. Groups of 5 oocytes at a time were incubated in 500 µl of extracellular solution for times indicated in individual figures, following which 100 µl was removed (for estimation of the extracellular concentration of radiolabel) and efflux immediately quenched by washing the oocytes with 4 × ice-cold ND96. The washed oocytes were transferred immediately to 96-well plates for estimation of the amount of radiolabel retained within the oocytes at the time of sampling.

To determine the efflux that was attributable to each of the two transporters of interest, the amounts of radioactivity measured in the extracellular medium and retained within the oocytes in the experiments with control $H_2O$-injected oocytes, were subtracted from those measured in *Tg*ApiAT6-1-expressing and *Tg*ApiAT1-expressing oocytes.

All oocyte uptake and efflux experiments were performed using either [$^{14}$C]- or [$^{3}$H]-labelled forms of the amino acid being studied, with the total substrate concentrations and concentration of [$^{14}$C]- or [$^{3}$H]-labelled amino acids specified in the relevant figure legends. The specific activities of [$^{14}$C]-labelled compounds were as follows: Lys, 301 µCi/mmol; Arg, 312 µCi/mmol; Met, 68 µCi/mmol; Leu, 328 µCi/mmol; Ile, 328 µCi/mmol; Gln, 253 µCi/mmol; Glu, 281 µCi/mmol; Asp, 200 µCi/mmol; Ala, 151 µCi/mmol; Pro, 171 µCi/mmol; and Gly, 107 µCi/mmol (Gly). For [$^{3}$H]-labelled compounds, specific activities were: putrescine, 28 Ci/mmol; spermidine, 32.35 Ci/mmol; and γ-[2-3-$^{3}$H(N)]GABA, 89 Ci/mmol. Following the relevant incubation periods, oocytes or aliquots of solutions were distributed into OptiPlate96-well plates (Perkin-Elmer) and oocytes were lysed overnight in 10% (w/v) SDS. Microscint-40 scintillation fluid (Perkin-Elmer) was added to the samples, and plates covered and shaken for 5 min before radioactivity was counted on a Perkin-Elmer MicroBeta$^{2}$ 2450 microplate scintillation counter.

## Oocyte protein detection assays

Oocyte surface biotinylation and whole membrane preparations were performed as described previously [63,66], using equivalent numbers of oocytes per experiment. During surface biotinylation, 15 oocytes were selected 4–6 days post cRNA injection, washed 3 × in ice-cold PBS (pH 8.0), and incubated for 25 mins at ambient temperature in 0.5 mg/ml of EZ-Link Sulfo-NHS-LC-Biotin (Thermo Fisher Scientific). To quench biotinylation, oocytes were washed 3 × in ice-cold PBS and then solubilised in lysis buffer (20 mM Tris-HCl pH 7.6, 150 mM NaCl, 1% v/v Triton X-100) on ice for 1.5 hr. To remove unsolubilised material, samples were centrifuged at $16,000 × g$, with the supernatant retained. To purify biotinylated proteins, the supernatant mixed was mixed with streptavidin-coated agarose beads (Thermo Fisher Scientific). The mixture was incubated with slow rotation for >2 hr before the beads were washed four times with lysis buffer then dissolved in SDS-PAGE sample buffer. For whole cell membranes, 25 oocytes were triturated in homogenisation buffer (50 mM Tris-HCl pH 7.4, 100 mM NaCl, 1 mM EDTA, protease inhibitors) and the resulting homogenate centrifuged at $2,000 × g$ at 4˚C for 10 min to remove nuclei and cell debris, then at $300,000 × g$ for 1 hr at 4˚C to pellet membranes. Samples were washed 3 × with homogenisation buffer and then solubilised with the same buffer containing 4% (w/v) SDS, and then prepared for SDS-PAGE.

## Oocyte metabolite extraction and LC-MS/MS data acquisition

*Xenopus laevis* oocytes injected with either *Tg*ApiAT1 cRNA, *Tg*ApiAT6-1 cRNA or $H_2O$ were incubated with substrate at concentrations, pH and temperatures indicated in figure legends.

All incubations occurred in substrates solved in $1 \times$ ND96 buffer as described previously [36]. For each condition, 12 oocytes were washed $\times 3$ with $1 \times$ ND96 solution not containing substrate under ambient conditions, before addition of $1 \times$ ND96 containing substrate. Oocytes requiring the replacement of one incubation solution with another (*e.g.* to examine efflux following pre-loading) were washed $\times 4$ in the ice-cold replacement $1 \times$ ND96 solution (minus substrate) before the addition of $1 \times$ ND96 containing substrate at ambient temperature. Oocyte incubations were quenched by placing oocyte-containing tubes on ice and washing $\times 4$ with 1 ml of ice-cold MilliQ $H_2O$. Polar metabolites were extracted using a two-stage liquid-liquid phase extraction. The first extraction was in chloroform:water:methanol (1:1:3) to isolate aqueous metabolites. Oocytes were lysed and titurated in this mixture and then centrifuged at $13,000 \times g$ for 5 min to remove cell debris and unsolved material. The second extraction involved adding 1:5 $H_2O$:mixture, which precipitated hydrophobic solutes. The supernatant was centrifuged again at $13,000 \times g$ for 5 min, which left 3 clear phases: a floating aqueous phase of water/methanol, a dense organic phase and an interphase containing a clear white precipitate. The upper aqueous phase was removed and the organic phase and interphase discarded. Samples were desiccated on a vacuum centrifuge, then re-solved with acetonitrile/$H_2O$ (80%/20% v/v).

Chromatographic separation was performed on an Ultimate 3000 RSLC nano Ultra high performance liquid chromatography (UHPLC) system (Dionex) by using hydrophilic interaction ion chromatography with a ZIC cHILIC column (3.0 μm polymeric, 2.1x150mm; Sequant, Merck) as described previously [36]. The gradient started with 80% mobile phase B (Acetonitrile; 0.1% v/v Formic acid) and 20% mobile phase A (10mM ammonium formate; 0.1% v/v formic acid) at a flow rate of 300 μl/min, followed by a linear gradient to 20% mobile phase B over 18 min. A re-equilibration phase of 12 mins using 80% mobile phase B was done with the same flow rate, making a total run time of 30 mins. The column was maintained at 40°C and the injection sample volume was 4μl. The mass detection was carried out by Q-Exactive Plus Orbitrap mass spectrometer (Thermo Scientific, Waltham, MA, USA) in positive electrospray mode. The following settings were used for the full scan MS: resolution 70,00, m/z range 60–900, AGC target $3 \times 10^6$ counts, sheath gas 40 l min$^{-1}$, auxiliary gas 10 l min$^{-1}$, sweep gas 2 l min$^{-1}$, and capillary temperature 250°C, spray voltage +3.5kV. The MS/MS data was collected through data dependent top 5 scan mode using high-energy c-trap dissociation (HCD) with resolution 17,500, AGC target $1 \times 10^5$ counts and normalized collision energy (NCE) 30. The rest of the specifications for the mass spectrometer remained unchanged from the vendor recommended settings. A pooled sample of all extracts was used as a quality control (QC) sample to monitor signal reproducibility and stability of analytes. Blank samples and QC samples were run before and after the batch and QC samples were run within the batch to ensure reproducibility of the data. Arg and Lys were calibrated as an external standard by first making serial concentration and then aliquoting the same volume from each (8 μl) to give the following amounts (pmol): 80, 200, 400, 800, 2000, 4800. Calibration curves for both metabolites were freshly determined with each new batch of LC-MS/MS run. The acquired raw metabolite data were converted into mzXML format and processed through an open source software MS-DIAL [69]. Identification of metabolites was performed by first using publicly available MS/MS libraries matching exact mass (MS tolerance 0.01 Da) and mass fragmentation pattern (MS tolerance 0.05 Da), and then further confirmed by standards through retention time where possible. Raw peak height was used for the quantification of metabolites.

### Electrophysiological recordings in *X. laevis* oocytes

Single oocytes were recorded in either unclamped mode to record membrane potential ($E_m$) or in two-voltage clamp configuration at a set membrane potential to record membrane

currents. Perfusion of different buffers and substrate solutions was controlled by valve release and stop, and perfusion rate either gravity-fed or controlled by a peristaltic pump (Gilson, Middleton, WI, U.S.A.). The $E_m$ of oocytes was recorded in unclamped mode using conventional boronsilicate glass microelectrodes capillaries (World Precision Instruments) filled with 3 M KCl to a resistance of 5–10 MΩ. In two-voltage clamp configuration, the same experimental setup was followed with the exception that borosilicate glass microelectrodes were filled with 3M KCl with a tip resistance of: $1.5 \geq R_e \geq 0.5$ MΩ. Oocytes were impaled and allowed to recover for 10 mins under constant perfusion to a steady-state $E_m$ before recordings began. All $E_m$ recordings were conducted in ND96 (pH 7.3) buffer. The amplifier was placed in set-up (current clamp) mode and the oocytes impaled with both the voltage sensing and current passing microelectrode. Before voltage clamping, the amplifier output current was set to zero to normalise currents recorded in voltage clamp mode. A test membrane potential pulse was also routinely administered and current output adjusted using amplifier gain and oscillation control (clamp stability), until the response time was sufficiently rapid (*i.e.* replicating the square wave-form with less than 5 $\mu$s maximal response time). Oocytes were clamped at −50 mV unless indicated otherwise.

All $E_m$ and membrane current recordings were made with voltage commands generated using a Axon GeneClamp 500B amplifier (Axon Instruments, Union City, CA, U.S.A.) connected to 1×LU and 10×MGU head stages. Amplifier gain was set at × 10. All output signals were low-pass filtered at 1 kHz. The analogue signal was converted into digital by a Digidata 1322A (Axon Instruments), and data were sampled at 3–10 Hz using Clampex v 10.0 software (Axon Instruments). Various buffers of different salt composition were utilised during free voltage and two-voltage clamp recordings, the composition of which are provided in S3 Table.

## Data analysis and statistics

Data analyses for the radiolabelled uptake experiments in parasites were performed using GraphPad Prism (Version 8). All oocyte data were analyzed using OriginPro (2015). All data displayed in figures represent the mean ± S.D. except where otherwise indicated. Unless uptake data from uninjected oocytes is included in figures, uptake in uninjected oocytes was subtracted from uptake in cRNA-injected oocytes to give the 'transporter-dependent uptake'. All data sets assumed Gaussian normalcy which was tested by running a Shapiro-Wilk test prior to analysis. Normalcy was determined as at the $P < 0.05$ threshold. Multi-variant experiments with three or more experimental conditions were subjected to a one-way ANOVA with Dunnett's post-hoc test and significance tested at the $p < 0.05$ level.

The setting of initial rate membrane transport conditions were established by fitting time-course uptake data in oocytes to:

$$S_t = S_{max}(1 - e^{-kt}) \tag{1}$$

which is a Box Lucas 1 model with zero offset [70], where $S_t$, and $S_0$, and are the amount of substrate (S) at variable time (t), or when $t = 0$, $S_{max}$ is the vertical asymptote of substrate amount, and k is the 1st order rate constant.

The theoretical reversal membrane potential ($E_{rev}$) for monovalent cationic amino acids ($AA^+$) in oocytes (S9D Fig) was determined using the Nernst equation:

$$E_{rev} = \frac{RT}{zF} \ln \frac{[AA^+]_{outside}}{[AA^+]_{inside}} \tag{2}$$

Where, R, T and F have their usual values and meanings, $z = +1$ for the $AA^+$ charge.

Calibration curves for LC-MS/MS quantification of Lys and Arg concentration in oocytes were fitted to a linear equation. Likewise, Lineweaver-Burke linear regressions of Michaelis-Menten steady-state kinetic data were also fitted to linear equations. All curve fittings were evaluated using adjusted goodness of fit $R^2$ values as quoted in figure legends. All non-linear fitting was conducted using the Levenburg-Marquardt algorithm, with iteration numbers varying from 4 to 11 before convergence was attained.

## Supporting information

**S1 Fig. Generating an ATc-regulated *Tg*ApiAT6-1 strain. A**. Schematic depicting the promoter replacement strategy to generate the ATc-regulated *Tg*ApiAT6-1 strain (r*Tg*ApiAT6-1), and the positions of screening primers used in subsequent experiments to validate successful promoter replacement. The native locus (top) and promoter-replaced locus (bottom) are shown. DHFR^PyrR, pyrimethamine-resistant dihydrofolate reductase cassette; t7s4, ATc-regulatable teto7-sag4 promoter. The teto7-sag4 promoter is bound by a tetracycline-controlled transactivator protein that facilitates transcription of the downstream gene (*Tg*ApiAT6-1 in this instance). The addition of the tetracycline analogue ATc to the culture medium results in binding of ATc to the transactivator protein, which inhibits binding of the transactivator protein to the teto7-sag4 promoter, and consequently reduces transcription of the downstream gene [71]. **B-E**. PCR analysis using genomic DNA extracted from native RH strain (WT) and modified r*Tg*ApiAT6-1 strain parasites, with primers that specifically detect the 3' region of the native locus (B), the 3' region of the modified locus (C), the 5' region of the native locus (D), and the 5' region of the modified locus (E).
(TIF)

**S2 Fig. Optimization of *Tg*ApiAT6-1 expression and activity in *X. laevis* oocytes. A**. Western blot with anti-HA antibodies to detect proteins from surface-biotinylated and total membrane fractions of oocytes injected with *Tg*ApiAT6-1 cRNA or uninjected (U.I.) controls. Each lane contains protein equivalents from equal oocyte numbers. **B**. Time-course measuring Lys uptake in *Tg*ApiAT6-1-expressing in oocytes one to five days post-cRNA injection. Each data point represents the mean ± S.D. uptake of 10 oocytes for a single experiment, and is representative of three independent experiments. Uptake was measured in the presence of 100 µM unlabelled Lys and 1.0 µCi/ml [14C]Lys. The uptake of Lys in uninjected oocytes has been subtracted for all days post-cRNA injection tested. **C**. Lys uptake in *Tg*ApiAT6-1-expressing in oocytes injected with 0–50 ng of *Tg*ApiAT6-1-encoding cRNA. Each data point represents the mean ± S.D. uptake of 10 oocytes for a single experiment and are representative of three independent experiments. Uptake was measured in the presence of 100 µM unlabelled Lys and 1.0 µCi/ml [14C]Lys. All measurements were conducted on day 4 post-cRNA injection. The uptake of Lys in a single uninjected oocyte batch has been subtracted from all data points. **D**. Time-course of Lys (open squares) and Arg (closed squares) uptake into *Tg*ApiAT6-1 expressing oocytes for determination of initial rate conditions. Uptake was measured in the presence of 100 µM unlabelled Arg and 1.0 µCi/ml [14C]Arg or 100 µM unlabelled Lys and 1.0 µCi/ml [14C]Lys. Each data point represents the mean ± S.D. uptake of 10 oocytes for a single experiment, and is representative of three independent experiments. The uptake of Lys or Arg at the same concentration in uninjected oocytes was subtracted from all data points.
(TIF)

**S3 Fig. Transmembrane flux activity in *X. laevis* control oocytes not expressing *Tg*ApiAT6-1 and transmembrane Lys efflux and retention in oocytes expressing *Tg*ApiAT6-1. A.** Uptake of a range of amino acids into oocytes not expressing *Tg*ApiAT6-1. Uptake was

measured in the presence of 100 μM unlabelled substrate and 1.0 μCi/ml [³H] or [¹⁴C] substrate. Amino acid substrates are represented by single letter codes, while for other metabolites: Pu, putrescine; Sp, spermidine; and GA, γ-amino butyric acid (GABA). Each bar represents the mean ± S.D. uptake of 10 oocytes for a single experiment paired with *Tg*ApiAT6-1 expressing oocytes from Fig 2B, and each is representative of three independent experiments. **B-C**. Uptake of 100 μM unlabelled Arg and 1.0 μCi/ml [¹⁴C]Arg was measured in oocytes not expressing *Tg*ApiAT6-1 in the presence of 1 mM (B) or 10 mM (C) of the competing amino acid. Amino acid substrates are represented by single letter codes. Each bar represents the mean ± S.D. uptake of 10 oocytes for a single experiment paired with *Tg*ApiAT6-1 expressing oocytes from Fig 2C and 2D, and are representative of three independent experiments. The first bar in each graph represents the Arg-only uptake control. **D-E**. Oocytes not expressing *Tg*ApiAT6-1 (injected with H₂O) were pre-loaded by microinjecting the indicated substrates to a final concentration of ~5 mM and the uptake of 15 $\mu$M Lys and 1.0 μCi/ml of [¹⁴C]Lys (D) or 1 mM Arg and 1.0 μCi/ml of [¹⁴C]Arg (E) was measured. Amino acid substrates are represented by single letter codes, while for other metabolites: Cr, creatine; Ag, agmatine; Sp, spermidine; Pu, putrescine; Ci, citrulline; and Or, ornithine. Each bar represents the mean ± S.D. uptake of 10 oocytes for a single experiment paired with *Tg*ApiAT6-1 expressing oocytes from Fig 5E and 5F, and are representative of three independent experiments. **F.** Lys efflux and retention in *Tg*ApiAT6-1 expressing oocytes in the presence of candidate *trans*-stimulating substrates. *Tg*ApiAT6-1-injected oocytes were pre-loaded over 8 hrs with 1 mM unlabelled Lys and 1.0 μCi/ml of [¹⁴C]Lys (*Tg*ApiAT6-1). Efflux of pre-loaded Lys (Lys$_o$; black bars) and retention of pre-loaded Lys (Lys$_i$; white bars) were measured after 1 hr (*Tg*ApiAT6-1) in the presence of 1 mM candidate counter substrates or with ND96 buffer in the extracellular medium (−). The horizontal dashed line across both figures indicates the amount of Lys pre-loaded (PL) into oocytes (left-most bar). Each bar represents the mean ± S.D. efflux or retention in from 3 batches of 5 oocytes from one experiment, one experimental repeat was conducted. Statistical analysis compares all bars to oocyte efflux in the presence of ND96 (−) (pH 7.3) (* $P < 0.05$, one-way ANOVA, Dunnett's post-hoc test). (TIF)

**S4 Fig. Electrical activity of *Tg*ApiAT6-1 is coupled to substrate translocation. A**. Paired representative membrane current recordings recorded from *Tg*ApiAT6-1 expressing oocytes superfused with 1 mM Arg in the presence of different extracellular salt compositions. The membrane current recordings were made under voltage-clamp configuration (E$_m$ = −50mV). Superfusion buffers indicated below tracings are: ND96 (Na⁺ buffer), KD96 (Na⁺ replaced by K⁺), Gluconate (Cl⁻ replaced by gluconate), ChCl96 (Na⁺ replaced by choline), ND96 −CaCl₂ (Ca²⁺ removed). All recordings were made at pH 7.3. Full buffer compositions are listed in S3 Table. **B**. Membrane current recordings in *Tg*ApiAT6-1 expressing oocytes as described in A for voltage-clamp recordings. Each bar represents the mean ± S.D. of inward currents from 8–12 oocytes per condition. Statistical analysis compares the mean of *Tg*ApiAT6-1 expressing oocytes with their buffer pair in H₂O-injected oocytes (*, $P < 0.05$, one-way ANOVA, Dunnett's post-hoc test). Note that all currents in A-B were recorded in two-voltage clamp configuration set to a membrane potential of −50 mV to record membrane current or in unclamped mode for recording membrane potential. **C**. The uptake of Arg in *Tg*ApiAT6-1 expressing oocytes incubated in different buffer compositions. Uptake was measured in the presence of 1 mM unlabelled Arg and 1.0 μCi/ml [¹⁴C]Arg. The removed salt from standard ND96 buffer (pH 7.3) is indicated below the graph. Na⁺-replacement salts are indicated above their respective bars. Each bar represents the mean ± S.D. uptake of 10 oocytes for a single experiment, and are representative of three independent experiments. Statistical analysis compares all bars

of *Tg*ApiAT6-1 expressing oocytes incubated in different buffer compositions ($P > 0.05$; one-way ANOVA, Dunnett's post-hoc test).
(TIF)

**S5 Fig. Time-courses of Lys, Arg and 2-DOG uptake in r*Tg*ApiAT6-1 and WT parasites.**
**A-D**. Uptake of Lys (A, C) or Arg (B, D) in r*Tg*ApiAT6-1 (A, B) or WT (C, D) parasites across a 20 min time-course. Lys uptake was measured in a solution containing 50 μM unlabelled Lys and 0.1 μCi/ml [$^{14}$C]Lys. Arg uptake was measured in a solution containing 80 μM unlabelled Arg and 0.1 μCi/ml [$^{14}$C]Arg. **E**. Uptake of 2-deoxyglucose (2-DOG) in r*Tg*ApiAT6-1 parasites across an 8 min time-course. 2-DOG uptake was measured in 25 μM unlabelled 2-DOG and 0.2 μCi/ml [$^{14}$C]2-DOG. All data points represent the mean ± S.D. from three independent experiments. One phase exponential curves were fitted to the data.
(TIF)

**S6 Fig. *Tg*ApiAT6-1 is important for parasite proliferation at a range of [Lys] and [Arg] in the culture medium.** Fluorescence growth assays measuring proliferation of r*Tg*ApiAT6-1/Tomato parasites in either the absence (black) or presence (red) of ATc. Parasites were cultured in RPMI medium containing a range of [Lys] (1–2500 μM) with [Arg] constant at 200 μM (**A**) or a range of [Arg] (1–2500 μM) with [Lys] constant at 200 μM (**B**). Parasite proliferation at each amino acid concentration was expressed as a percentage of proliferation of parasites cultured at the highest Lys or Arg concentration when these parasites were at mid-log stage growth. Data points represent the mean ± S.D. of three independent experiments, each consisting of three technical replicates.
(TIF)

**S7 Fig. The putative Lys biosynthesis pathway of *T. gondii* does not contribute to tachyzoite proliferation. A.** Scale diagram of the amino acid sequence of the *Tg*LysA protein, with the positions of predicted active site residues depicted in cyan, and the position of the frameshift mutation generated in the *lysA*$^{Δ281-68}$ strain depicted in red. **B.** Fluorescence growth assays measuring proliferation of WT (RH/Tomato; black) and *lysA*$^{Δ281-68}$ (red) parasites cultured in RPMI medium containing a range of [Lys] (3–800 μM). Parasite proliferation is expressed as a percentage of growth in mid-log stage parasites cultured in 400 μM Lys. Data points represent the mean ± S.D. of four independent experiments, each consisting of three technical replicates.
(TIF)

**S8 Fig. H$_2$O-injected oocytes do not efflux, and are not *trans*-stimulated by, cationic amino acids.** H$_2$O-injected oocytes were pre-loaded with either 1 mM unlabelled Lys and 1.0 μCi/ml of [$^{14}$C]Lys (A) or 1 mM unlabelled Arg and 1.0 μCi/ml of [$^{14}$C]Arg (B) until they had reach the same approximate intracellular concentration of radiolabelled substrate (21–28 hr) as *Tg*ApiAT6-1- and *Tg*ApiAT1-expressing oocytes (Fig 5A and 5B). The retention of substrates was measured in the presence of 1 mM external substrate (closed symbols) or in the absence of an external substrate (open symbols). Data points represent the mean ± S.D. from 3 batches of 5 oocytes from one experiment, and are representative of 3 independent experiments.
(TIF)

**S9 Fig. Net substrate transport and *trans*-stimulation specificity of *Tg*ApiAT6-1 and *Tg*ApiAT1. A-B**. LC-MS/MS calibration curves for Lys (A) and Arg (B). The linear regressions were fitted with an $R^2 = 0.99$ for both curves. **C**. The oocyte membrane potential (E$_m$) of *Tg*ApiAT6-1 expressing oocytes, *Tg*ApiAT1 expressing oocytes, H$_2$O-injected (H$_2$O-inj) control

oocytes and uninjected (U.I.) control oocytes were measured in unclamped mode. Resting $E_m$ measurements were conducted following incubation for 4 day post-cRNA injection in oocyte Ringer ($OR^{2+}$) buffer, with oocytes transferred to ND96 buffer for recording. Oocytes with $E_m$ < −23 mV were discarded from further analysis. Each bar represents the mean ± S.D. of inward currents with the number of oocytes recorded as follows: *Tg*ApiAT6-1 (*n* = 16), *Tg*ApiAT1 (*n* = 19), $H_2O$-injected (*n* = 13), and uninjected controls (*n* = 19). Statistical analysis compares all bars to uninjected controls (* *P* < 0.05, one-way ANOVA, Dunnett's post-hoc test). **D**. The theoretical equilibrium distribution ($[AA^+]_{inside}/[AA^+]_{outside}$) vs membrane potential ($E_m$) for a monovalent cation across a freely diffusible membrane as calculated by the Nernst equation (see Materials and Methods, Eq 2).
(TIF)

**S1 Table. Metabolite fold-change upon incubation of *Tg*ApiAT6-1-expressing oocytes in a solution containing 1 mM Lys for 25 hr.** Included are the average retention time (R.T.) of the metabolite, the average mass-to-charge (m/z) ratio of the ion, the name of the metabolite, and the quality control standard deviation (QC Relative S.D). The fold change of the substrate and other detected metabolites was determined by dividing the value at 25hrs with the 0 hr value for *Tg*ApAT6-1 expressing and uninjected (U.I.) oocytes.
(DOCX)

**S2 Table. Metabolite fold-change upon incubation of *Tg*ApiAT1- or *Tg*ApiAT6-1-expressing oocytes in a solution containing 1 mM Arg for 25 hr.** Included are the average retention time (R.T.) of the metabolite, the average mass-to-charge (m/z) ratio of the ion, the name of the metabolite, and the quality control standard deviation (QC Relative S.D). The fold change of the substrate and other detected metabolites was determined by dividing the value at 25hrs with the 0 hr value for *Tg*ApiAT1-expressing oocytes, *Tg*ApAT6-1 expressing oocytes and uninjected (U.I.) oocytes.
(DOCX)

**S3 Table. Solution composition used for uptake and electrophysiological recordings in *X. laevis* oocytes.**
(DOCX)

## Acknowledgments

We thank Angelika Bröer for many helpful suggestions and technical advice during the course of this project. We also thank Ben Durant and the RSB animal service team for animal husbandry, *Xenopus laevis* surgery and oocyte preparation. We are grateful to the students of the 2015 Biology of Parasitism Course (Marine Biological Laboratory, Woods Hole, MA) for early studies into the function of *Tg*ApiAT6-1. We thank Harpreet Vohra and Michael Devoy for performing cell sorting.

## Author Contributions

**Conceptualization:** Stephen J. Fairweather, Esther Rajendran, Kiaran Kirk, Stefan Bröer, Giel G. van Dooren.

**Data curation:** Stephen J. Fairweather, Martin Blume, Kiran Javed.

**Formal analysis:** Stephen J. Fairweather, Esther Rajendran, Martin Blume, Kiran Javed.

**Funding acquisition:** Malcolm J. McConville, Kiaran Kirk, Stefan Bröer, Giel G. van Dooren.

**Investigation:** Stephen J. Fairweather, Esther Rajendran, Martin Blume, Kiran Javed, Birte Steinhöfel, Giel G. van Dooren.

**Methodology:** Stephen J. Fairweather, Esther Rajendran, Martin Blume, Kiran Javed, Stefan Bröer, Giel G. van Dooren.

**Project administration:** Stephen J. Fairweather, Kiaran Kirk, Stefan Bröer, Giel G. van Dooren.

**Resources:** Malcolm J. McConville, Stefan Bröer, Giel G. van Dooren.

**Supervision:** Malcolm J. McConville, Kiaran Kirk, Stefan Bröer, Giel G. van Dooren.

**Validation:** Stephen J. Fairweather, Esther Rajendran, Martin Blume, Kiran Javed, Birte Steinhöfel.

**Visualization:** Stephen J. Fairweather, Esther Rajendran, Martin Blume, Kiran Javed, Giel G. van Dooren.

**Writing – original draft:** Stephen J. Fairweather, Giel G. van Dooren.

**Writing – review & editing:** Stephen J. Fairweather, Esther Rajendran, Martin Blume, Kiran Javed, Birte Steinhöfel, Malcolm J. McConville, Kiaran Kirk, Stefan Bröer, Giel G. van Dooren.

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
