## [Decision Letter · Decision Letter 0]

4 May 2021

Dear Dr. van Dooren,

Thank you very much for submitting your manuscript "Coordinated Action of Multiple Transporters in the Acquisition of Essential Cationic Amino Acids by the Intracellular Parasite Toxoplasma gondii" for consideration at PLOS Pathogens. As with all papers reviewed by the journal, your manuscript was reviewed by members of the editorial board and by several independent reviewers. In light of the reviews (below this email), we would like to invite the resubmission of a significantly-revised version that takes into account the reviewers' comments.

Overall the reviewers are enthusiastic about the work, considering it an an important contribution to the field and in principle suitable for publication in PLOS Pathogens. They have all raised some minors issues that should be easily fixed. In additionally, Reviewer #2 is requesting some additional experiments for completeness. I trust that you will find them straightforward to address and relevant to strengthen your findings.

We cannot make any decision about publication until we have seen the revised manuscript and your response to the reviewers' comments. Your revised manuscript is also likely to be sent to reviewers for further evaluation.

Sincerely,

Dominique Soldati-Favre

Associate Editor

PLOS Pathogens

Kami Kim

Section Editor

PLOS Pathogens

Kasturi Haldar

Editor-in-Chief

PLOS Pathogens

orcid.org/0000-0001-5065-158X

Michael Malim

Editor-in-Chief

PLOS Pathogens

orcid.org/0000-0002-7699-2064

Reviewer's Responses to Questions

**Part I - Summary**

Reviewer #1: Fairweather, et al. present a model of cationic amino acid transport by TgApiAT6-1 as well as TgApiAT1 in T. gondii parasites. The experiments conducted provide solid evidence that TgApiAT6-1 transports both Lys and Arg, with a preference for Lys transport. Overall, this is a well-conducted study with novel findings that will be of interest to parasitologists and researchers interested in solute transport in intracellular pathogens. However, I have a number of comments and suggestions for improvement.

Reviewer #2: This manuscript by Fairweather et al. presents the characterization at the cellular and physiologic levels of cationic amino acids in Toxoplasma gondii. Based on homology, the group has identified a large family of apicomplexan-specific amino acid transporters named ApiATs, including an arginine transporter-TgApiAT1. In the present study, the authors show that Toxoplasma encodes a second arginine transporter, TgApiAT6-1, that mediates primarily uptake of lysine and low-affinity uptake of arginine. The authors show that depleting TgApiAT6-1 using regulatable knockdown leads to a drastic decrease of parasite replication which is specific to loss of protein as complemented cells restored the WT phenotype. Based on amino acid uptake and electrophysiology measurements, the authors propose that the uptake of cationic amino acids by TgApiAT6-1 is ‘trans-stimulated’ by cationic and neutral amino acids and is likely promoted by inwardly negative membrane potential and with a high affinity for lysine.

The major advance made in this paper is showing that Toxoplasma encodes two arginine/lysine amino acid transporters suggesting that Toxoplasma evolved mechanisms of acquiring amino acids from the host cells to establish the infection. In addition, they show that Toxoplasma is auxotrophic for lysine as lower levels of lysine control parasite replication. The results presented in this manuscript added important aspects of amino acid acquisition during the replicative stage of infection. Unfortunately, the authors did not address the replication rate of rTgApiAT6-1 in different media such as DMEM or RPMI or complemented with arginine and/or lysine neither confirm the membrane localization of the transporter. In addition, the parasite replication in the mouse model is missing which would add the importance of this transporter to acute infection.

Reviewer #3: Fairweather and colleagues present a thorough and detailed study of essential cationic amino acid transporters in Toxoplasma gondii. They demonstrate TgApiAT6-1is essential for parasite propagation most likely through high affinity lysine uptake. They also present evidence TgApiAT6-1 transports arginine, with relatively low affinity (along with the previously reported TgApiAT1) and that cationic amino acid transport via TgApiAT6-1 is trans-stimulated by a range of cationic and neutral amino acids. In addition, the authors propose a model to explain the regulated uptake of cationic amino acids is different environments.

Strengths: This is a large and detailed study, using a range of molecular and functional methodologies to examine cationic amino acid transport in T. gondii. The text is easy to follow, the data look robust, with appropriate statistical analysis and the findings are novel and significantly improve our understanding to transport processes in apicomplexan parasites.

Weaknesses: These are few and I have a number of comments, mostly minor in nature, which could be considered by the authors to improve the manuscript (as noted below)

**Part II – Major Issues: Key Experiments Required for Acceptance**

Reviewer #1: No major issues - all are relatively minor.

Reviewer #2: 1. The proposed function of TgApiAT6-1 requires that it is found in the Toxoplasma membrane. While the localized is based on homology, this approach does not provide the necessary clearly evidence that TgApiAT6-1 localizes in the parasite membrane. It needs to be established that TgApiAT6-1 is indeed the membrane using a technique such as immunofluorescence using intra and extracellular parasites.

2. Given that TgApiAT6-1 depletion results in defects on arginine/lysine uptake and severe replication defect may mostly be related to the change in availability of intracellular amino acids. However, the current study does not establish the replication of rTgApiAT6-1 in presence of different concentrations of arginine and/or lysine. A more thorough investigation of parasite replication in TgApiAT6-1-depleted parasites, including a doubling assay/proliferation assay in RPMI and DMEM, may help establish why TgApiAT6-1 is essential.

3. In the proposed working model (figure 7), the authors speculate that in cells with low arginine concentration (e.g liver) TgApiAT6-1 would preferentially uptake lysine intend of arginine, while in cells with high arginine concentration (e.g. kidneys), arginine and lysine are uptake by TgApiAT6-1. If this is the case, the authors should show the parasite replication in kidneys cell line (HEK 293) and liver cell line (HepG2) in presence or absence of arginine and/or lysine to help understand the role of TgApiAT6-1 in acute infection. Also, it will help to understand the role of TgApiAT6-1 by infection of the animal model with WT, rTgApiAT6-1 and cTgApiAT6-1 parasite and analyze the animal mortality.

Reviewer #3: There are no issues that I would consider as major issues.

**Part III – Minor Issues: Editorial and Data Presentation Modifications**

Reviewer #1: Fairweather, et al. present a model of cationic amino acid transport by TgApiAT6-1 as well as TgApiAT1 in T. gondii parasites. The experiments conducted provide solid evidence that TgApiAT6-1 transports both Lys and Arg, with a preference for Lys transport. Overall, this is a well-conducted study with novel findings that will be of interest to parasitologists and researchers interested in solute transport in intracellular pathogens. However, I have a number of comments and suggestions for improvement.

Abstract: the authors should indicate that their experimental model was both with T. gondii parasites as well as transport studies in Xenopus laevis oocytes.

Figure 1: Panel A should include units for molecular weights (i.e. kDa). What concentration of aTC was used? Were different concentrations tested (i.e. titration)? Dose-response data, if available, should be included in the supplement.

Figure S1: Please explain in the text how this regulatable system works, in terms of the role of the teto-sag4 and how repression is achieved.

Figure 2: When describing the parasite amino acid data in panel A, the authors should add a note that Arg was not detected by GC-MS (it would help to also explain why). In Panels B-D, the background uninfected oocyte should be shown or listed somewhere in a supplemental table. There is no label on the first bar of Panel D. The x-axis label on Panel F should be [L-Arginine]. The labels on the axes are very small and hard to read. How did the authors select the concentrations of amino acids used in these assays?

It would be interesting to show the predicted topology of this transporter. Homology with other ApiAT proteins should be shown in the supplement.

Table 1: Vmax levels should be shown. Also, please clarify the meaning of K0.5 in the footnote.

Figure 3: In Panels A-C, the short horizontal lines indicating the drug pulse are not clear – maybe arrows or vertical lines should be added? In Panel E, Arginine is spelled out when it is abbreviated everywhere else. For Panels E-F, the length and timing of Arg pulse was not indicated. Also, the error bars in panel E are large. How can the authors be confident that the Michaelis-Menten fit is correct?

Figure 4: It would be interesting to do Lys and Arg competition assays in the knockdown parasite, varying one amino acid concentration to test its ability to inhibit transport of the other radiolabeled amino acid (at a fixed concentration) to determine the relative contribution of this transporter to Lys and Arg transport. This is not essential and have technical caveats but it would be useful to add if the data are available.

Figure S4: Transport does not appear to plateau at 20 min, especially with Arg. It would have been preferable to conduct these studies over a longer time frame.

Figure S5: A diagram of the TgLysA locus and the location of the catalytic domain would be useful. The authors mention that proliferation is inhibited below 50 uM, but there are so many points on the graph below 200 uM [Lys] that it is difficult to see this on the graph. Please add more minor ticks.

Figure 5: It is unclear why the experiment done in Panel A with TgApiAT6-1 and Lys was not also done with Arg – can the authors clarify. For Panels C-D, the Y-axis should be labeled Arg efflux to clarify. Experiments done in Panel C with TgApiAT6-1 should also preferably be done with Lys. Panel F is similar to Panel 2C-D – this should be spelled out and experimental differences between those data stated.

Figure 6: 50-54 hours was much longer than the other experiments. Why use this time course in this experiment and not others? Also, in the parasite both Arg transporters are indicated to be present on the parasite plasma membrane. Have the authors examined relative expression levels of these two proteins in the parasite?

Figure 7: This model fits with the conclusions of the paper. Is the Em of oocytes tested in this study consistent with that of parasites? It would be beneficial to test whether a membrane potential is important for transport.

Points to consider for the discussion:

Could these transporters be a drug target? It would be interesting to explore this concept and discuss whether selective inhibitors could be screened for. Also, do these ApiAT proteins have any naturally occurring mutations?

Typographical comments:

Pg. 13: Arg written as Arg+

Pg. 16: “twofold” to “two-fold”; keep consistency with writing out numbers or not

Pg. 17: “three-fold”

Figure 5: In the figure legend, “IUPAC 3-latter codes” should be “IUPAC 3-letter codes.

References are missing italics.

Figure 6: The legend has a typo: “A. A-C.”

Reviewer #2: Fig. 1B: Absence of TgApiAT6-1 decreases parasite proliferation, however after 3-4 days of ATc treatment the parasites are still viable? Can rTgApiAT6-1+ATc infect new cells?

Fig. 1D: What is the expression level of cTgApiAT6-1? Is it similar to WT levels? Given that cTgApiAT6-1 replicates faster than WT according to Fig. 1D.

Given that TgApiAT1 is upregulated in arginine limitation. What is the expression TgApiAT6-1 levels in parasite cultivate in lys/arg limitation?

Reviewer #3: 1. Lines 152 to 155. Was there a rationale for using 2 mg/ml labelled amino acids and a 15 min incubation time?

2. Line 158-159. The authors state other amino acids (other than lysine) could be transported by TgApiAT6-1 that are not detected by GC-MS. Another explanation is that the conditions used do not allow the detection of transport of other amino acids (this is covered 2 paragraphs later but should at least be noted here).

3. Line 257-276. The authors genetically disrupt TgLysA which has no effect on parasite proliferation over a range of lysine concentrations compared with wild type parasites. While I agree that the disruption would most likely render the truncated enzyme non-functional, there is no proof (phenotypic or functional) that this is the case, as the enzyme would seem to be non-essential. Perhaps, at the very least, the authors could note that this is consistent with previous CRISPR studies (ref [33]?), if that is the case (apologies if I missed this in the text somewhere).

4. Discussion/Figure 7. The authors provide a model for cationic amino acid uptake into T gondii-infected cells, using the kidney and the liver as examples of differing cationic amino acid environments. In the model, host CAT1 is noted (based on a study reporting infection induced upregulation of CAT1 but not CAT2 or 3 in a range of cells). However, I’m not sure liver cells were used in that study. Liver cells express CAT2A/B as the predominant cationic amino acid transporters. These have been shown to be important for liver stage malarial parasite development (Meireles et al 2017; PMID: 28642498). Perhaps this could be noted.

5. Methods. Please note the specific activity of the radiolabels used.

6. Line 670/S2 Fig. A. For the oocyte protein detection assays, was each well loaded with equal amounts of protein and how much?

7. Figures in general. I reviewed the manuscript printed on A4 paper and was unable to read many of the labels (particularly the amino acid abbreviations).

8. Figure 5 legend and possibly other places. The authors note in several places that “microinjecting 5 mM of Arg”. Perhaps this could be rephrased to “microinjecting Arg to a concentration of 5 mM” as it reads currently as though you are injecting a solution of 5 mM of Arg.

9. Figure S2 legend. For the description of unlabelled and labelled Arg and Lys 100 microM seems to mentioned too many times.

10. General. The authors often state that transport values in negative controls have been subtracted from the data presented in the figures. If possible, please provide an estimate of these transport measurements so as to give an idea of signal to noise.

11. Typos ("" add/[] remove): Line 49, concentrations “are” comparatively. Line 52, able “to” survive. Line 108, important for [the] tachyzoite. Fig 3 legend, line 11, and at [of] one

PLOS authors have the option to publish the peer review history of their article (what does this mean?). If published, this will include your full peer review and any attached files.

Reviewer #1: No

Reviewer #2: No

Reviewer #3: **Yes: **Dr Henry M. Staines
---

## [Editor Report · Decision Letter 1]

23 Jul 2021

Dear Dr. van Dooren,

We are pleased to inform you that your manuscript 'Coordinated Action of Multiple Transporters in the Acquisition of Essential Cationic Amino Acids by the Intracellular Parasite Toxoplasma gondii' has been provisionally accepted for publication in PLOS Pathogens.

Best regards,

Dominique Soldati-Favre

Section Editor

PLOS Pathogens

Kami Kim

Section Editor

PLOS Pathogens

Kasturi Haldar

Editor-in-Chief

PLOS Pathogens

orcid.org/0000-0001-5065-158X

Michael Malim

Editor-in-Chief

PLOS Pathogens

orcid.org/0000-0002-7699-2064
---

## [Editor Report · Acceptance letter]

19 Aug 2021

Dear Dr. van Dooren,

We are delighted to inform you that your manuscript, "Coordinated Action of Multiple Transporters in the Acquisition of Essential Cationic Amino Acids by the Intracellular Parasite <i>Toxoplasma gondii<i>," has been formally accepted for publication in PLOS Pathogens.

Best regards,

Kasturi Haldar

Editor-in-Chief

PLOS Pathogens

orcid.org/0000-0001-5065-158X

Michael Malim

Editor-in-Chief

PLOS Pathogens

orcid.org/0000-0002-7699-2064